# Acoustic frequency atomic spin oscillator in the quantum regime

Jun Jia[1], Valeriy Novikov [1,2], Tulio Brito Brasil [1], Emil Zeuthen [1], Jörg Helge Müller[1] & Eugene S. Polzik [1]✉

Quantum noise reduction and entanglement-enhanced sensing in the acoustic frequency range is an outstanding challenge relevant for a number of applications including magnetometry and broadband noise reduction in gravitational wave detectors. Here we experimentally demonstrate quantum behavior of a macroscopic atomic spin oscillator in the acoustic frequency range. Quantum back-action of the spin measurement, ponderomotive squeezing of light, and virtual spring softening are observed at oscillation frequencies down to the sub-kHz range. Quantum noise sources characteristic of spin oscillators operating in the near-DC frequency range are identified and means for their mitigation are presented.

Quantum mechanics implies that the measurement of a specific observable, e.g., position or a spin projection, is accompanied by the injection of noise in the canonically conjugate variable, e.g., momentum or another spin projection. This noise, resulting from quantum back-action (QBA)[1], together with the imprecision noise (shot noise), determines the precision bounds in quantum metrology tasks. The performance achieved with balanced (and uncorrelated) QBA and imprecision noise sources is referred to as the standard quantum limit (SQL). The microscopic mechanism behind the QBA depends on the physical platform. In the case of interferometric displacement measurements (such as in gravitational wave detectors), it is due to the shot noise of light, and manifests itself as fluctuations in the laser radiation-pressure force[2]. In spin-polarized systems the QBA mechanism is attributable to the light shift caused by quantum fluctuations of the Faraday probe polarization[3]. Recently, QBA has been observed in various quantum systems[4–6].

Atomic spin ensembles have become a rich resource for quantum sensing and for engineering macroscopic quantum states with applications in ultra-sensitive magnetometry, search for new physics, and interferometry[7–12]. A remarkable feature of spin ensembles is the ability to implement an effective negative-mass oscillator, demonstrated in several protocols, such as entanglement-assisted magnetometry[13,14] and quantum memory for a set of two-mode-squeezed states[15]. A central application of such an oscillator is the broadband QBA evasion in hybrid systems proposed in Refs. 16,17.

To date, quantum sensing beyond the SQL based on atomic spins has been predominantly performed in the MHz frequency range. QBA-free sensing in the acoustic frequency range would enable new sensing applications beyond the SQL. It has also become increasingly important in current and future gravitational wave detectors (GWDs)[18] as they approach SQL-limited performance in the acoustic frequency band[19–21]. As proposed in Refs. 22,23, combining a GWD with a negative-mass spin oscillator with the help of a recently demonstrated two-color source of entangled light[24] allows for cancellation of both shot noise and QBA noise, enabling broadband sensitivity beyond the SQL.

Here we demonstrate the QBA-limited performance of a spin oscillator in the audio-frequency band. Analogously to optomechanics[25], the spin ensemble can generate ponderomotive squeezing of light, i.e., reduction of noise via correlations between amplitude and phase quadrature fluctuations. We demonstrate ponderomotive squeezing tunable in its frequency down to 700 Hz. The correlations between the light quadratures also lead to another crucial element of low-frequency sensing that we present here: the virtual oscillator-frequency downshift, which is, for example, necessary for matching the spin response to that of a GWD[23] as well as for other sensing applications in the acoustic frequency range[26]. Furthermore, we observe and model the residual low-frequency noise sources limiting the present performance and outline ways to overcome them.

## Results
### Theoretical basis
A spin-polarized atomic ensemble precessing at frequency $\Omega_S \propto |\boldsymbol{B}|$ in a magnetic field $\boldsymbol{B}$ acts as an oscillator with an effective positive or negative mass depending on the orientation of the collective spin $\hat{\boldsymbol{J}}$

[1]Niels Bohr Institute, University of Copenhagen, Copenhagen, Denmark. [2]Russian Quantum Center, Skolkovo, Moscow, Russia. ✉e-mail: polzik@nbi.ku.dk

with respect to $\boldsymbol{B}$[17]. The ensemble is probed by light (Fig. 1) with the interaction defined by the quantum nondemolition (QND) Hamiltonian $\hat{H}_{int} \propto a_1 \hat{S}_z \hat{J}_z$[27], where $a_1$ is the vector polarizability and $\hat{S}_z$ is a component of the Stokes vector operator $\hat{\boldsymbol{S}}$[6]. The collective spin state is read out by measuring the quadrature of the probe optical field $\hat{Q}_L(\phi) = \hat{P}_L \cos(\phi) + \hat{X}_L \sin(\phi)$, where $\phi$ is the homodyne phase and $\hat{X}_L$ ($\hat{P}_L$) are the normalized Stokes operators $\hat{S}_z$ ($\hat{S}_y$) representing the amplitude (phase) quadrature, respectively. The power spectral density (PSD) $S_S$ for the detected optical field normalized to the shot noise is[28,29]

$$S_S(\Omega)|_{\hat{Q}_L(\phi)} = 1 + 4\eta S_{QBAN}\cos^2(\phi) + 2\eta S_{corr}\sin(2\phi) + 4\eta S_{TN}\cos^2(\phi) + \eta S_{bb}\cos^2(\phi). \tag{1}$$

The terms in Eq. (1) are the contributions from imprecision shot noise (SN), QBA noise (QBAN), cross-correlations between the QBAN and SN, atomic thermal fluctuations (thermal noise, TN), and broadband spin-response noise. The nominal imprecision noise level is represented by unity, the strength of the QBA noise term $S_{QBAN} = \Gamma_S^2 |\chi_S(\Omega)|^2$ is defined by the atomic readout rate $\Gamma_S \propto g_{cs}^2 S_x J_x \propto d$, where $g_{cs}$ is the photon-atom coupling rate and $d$ is the optical depth of the spin ensemble[27,28,30]. The spectral response of the oscillator is governed by the susceptibility function $\chi_S(\Omega) = \Omega_S/[(\gamma_S/2 - i\Omega)^2 + \Omega_S^2]$, where the spin damping rate $\gamma_S = \gamma_{S,0} + \gamma_{S,pb}$ is decomposed into a probe power-broadening part $\gamma_{S,pb} \propto \Gamma_S$ and an intrinsic linewidth $\gamma_{S,0}$. The term containing the correlations between QBAN and SN, $S_{corr} = \Gamma_S \text{Re}\left[\chi_S(\Omega)\right]$, present at $\phi \neq 0, \pi/2$, induces an effective frequency downshift (virtual spring softening) of the spin response to external forces *as it appears in the light field*[23], whose effect on the observed spectrum is discussed in the Results section. It is analogous to the virtual rigidity effect in quantum optomechanics[18].

The term $S_{TN} \approx 2\gamma_S \Gamma_S |\chi_S(\Omega)|^2 S_\zeta$ in Eq. (1) is the response of the spin oscillator to the stochastic force $\hat{\zeta}$ that has the spectrum $S_\zeta = (n_S + 1/2)$, where $n_S$ is the thermal occupancy of the spin oscillator. Finally, the contribution of $S_{bb}$ arises from extraneous, fast-decaying atomic modes coupling to the probe light[31]. In the present work, it is minimized by employing a top-hat probe beam with a high cell filling factor (Methods, Sec. B). The measurement precision of the indicated noise contributions except the nominal shot noise can be improved with a better overall detection efficiency $\eta$.

A proper choice of $\phi$ allows for destructive interference between SN and QBAN. As a result, the output light noise drops below the shot noise level in a certain frequency range, provided that the thermal contribution $\propto S_{TN}$ is sufficiently small. Besides its practical utility in various applications, such ponderomotive squeezing[25] allows us to calibrate the QBAN as discussed below. Analogously to the ponderomotive squeezing in optomechanics[32], the maximal degree of squeezing induced by the atomic ensemble in the limit of $\gamma_S \ll \Gamma_S, \Omega_S$ is

$$S_S(\Omega_{opt})|_{\hat{Q}_L(\phi_{opt})} \approx 1 - \eta \frac{C_q}{C_q + 1}, \tag{2}$$

and is achieved in a narrow frequency range around $\Omega \approx \Omega_{opt}$ when the optimal phase $\phi_{opt}$ of the detection quadrature is selected and the broadband noise is ignored. The quantum cooperativity

$$C_q = \frac{S_{QBAN}}{S_{TN}} = \frac{\Gamma_S}{\gamma_S(1 + 2n_S)}, \tag{3}$$

is the ratio between the QBAN and the thermal noise.

## Experimental setup

The ensemble of $N_S \approx 10^{10} - 10^{11}$ Cesium-133 atoms is contained in an antirelaxation-coated vapor cell ($2 \times 2 \times 80$ mm³) heated by a low-noise heater to 40 °C providing a large optical depth and cooperativity[31] (Fig. 1). To minimize the optical losses, both input and output surfaces are anti-reflection coated with an overall transmission of 96%. The PSD of the output probe light (~1 mW) is measured by polarization homodyne detection[15,27] with an overall detection efficiency of $\eta \approx 92\%$ and more than 14 dB shot noise clearance above the electronic noise for analysis frequencies down to 100 Hz. The homodyne phase $\phi$ is controlled by wave plates.

The spin oscillator is prepared by optical pumping of the atomic ensemble either to the lowest ($|F = 4, m_F = -4\rangle$) or to the highest ($|F = 4, m_F = 4\rangle$) Zeeman sublevel with a degree of spin polarization of $\lesssim 98\%$ (Methods, Sec. *Preparation and characterization of atomic state*). Low electro-magnetic noise, as required to reach quantum-limited performance, is achieved by a combination of magnetic coils operated with ultra low current noise and magnetic shielding (Methods, Sec. *Atomic vapor cell and PCB coils*). The widely tunable resonance frequency $\Omega_S$ of the spin oscillator is controlled by the

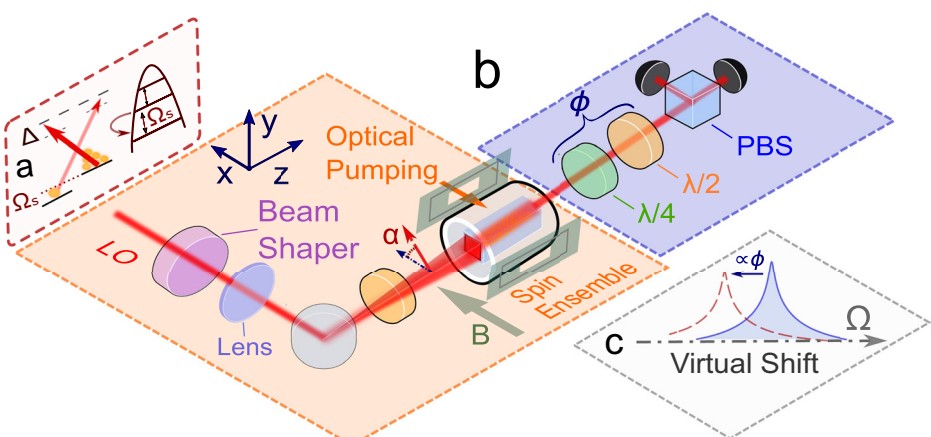

**Fig. 1 | Schematics of the experimental setup. a** The spin ensemble is probed by linearly polarized off-resonant light with a top-hat spatial mode shape. The probe polarization angle $\alpha$ with respect to the $x$-axis is adjusted for the QND measurement (Methods, Sec. *Spin alignment noise*) of the collective atomic spin. A quarter- and a half-wave plate define the quadrature phase $\phi$ detected by the polarization homodyning. **b** When prepared in highly polarized (coherent spin) state, the atomic ensemble can be described as two-level system, thus exhibiting the behavior of a harmonic oscillator. Specifically, we can prepare the atomic oscillator with the effective negative mass, creating inverted spin population. **c** The effect of ponderomotive squeezing, originating from cross-correlations between QBAN and SN, can be interpreted as a virtual shift of the resonance frequency.

magnitude of the applied magnetic field $\boldsymbol{B}$, scaling as 0.35 MHz/G. The sign of the effective oscillator frequency $\Omega_S$, equivalent to the sign of the effective mass, can be set by the direction of $\boldsymbol{B}$ or, alternatively, by the direction of circular polarization of the pump fields. The probe beam is linearly polarized at an angle $\alpha$ relative to the magnetization axis $x$. The frequency detuning $\Delta$ of the optical field from the transition $6S_{1/2}, F = 4 \leftrightarrow 6P_{3/2}, F' = 5$ is adjustable and was initially set to 1.6 GHz (see Methods, Sec. *Preparation and characterization of atomic state*).

From the analysis of the spin noise spectrum, we extract the parameters of the collective spin oscillator system appearing in Eq. (1); cross-validations of the readout rate $\Gamma_S$ are performed using the coherent induced Faraday rotation technique (CIFAR, see Ref. 33 and Methods, Sec. *Calibration of readout rate*). The thermal occupancy is found from the atomic spin polarization using the magneto-optical resonance method (MORS,[34] Sec. *Preparation and characterization of atomic state*). The reconstructed distribution of Zeeman sublevel populations allows for distinguishing between the positive- and negative-mass configurations (see Methods, Sec. *Spin noise spectra with effective masses*).

## Virtual frequency downshift of the observed spin oscillator response

We begin with characterization of the system in the upper part of the acoustic spectral range, setting the Larmor frequency $|\Omega_S|/(2\pi) = 18$ kHz. Importantly, we explore the configuration of an effective negative mass for the spin oscillator. Performing the fits of the spin noise spectra at phase quadrature $\hat{P}_L$ and the quadrature $\hat{Q}_L(\phi_{opt})$ yielding the strongest ponderomotive squeezing (Fig. 2a, b), we extract the essential parameters of the atomic spin ensemble. The readout rate $\Gamma_S/(2\pi) = 3.8$ kHz is in reasonable agreement with the results of the CIFAR calibration, whereas the amount of thermal noise, encoded in the thermal occupation $n_S = 3.5$, is larger than the value $n_S \approx 0.6$ obtained from MORS. This is likely due to noise sources not accounted for in the model of Eq. (1), for example, the ubiquitous intensity fluctuations of the probe laser, that are absent in MHz frequency range, but grow significantly toward the audioband. Consequently, we estimate the cooperativity $C_q \approx 3$. QBAN-dominated spin dynamics (Fig. 2a) is further confirmed by observation of strong ponderomotive squeezing $S_{SS} \lesssim -5.0$ dB (Fig. 2b). This value matches well the retrieved $C_q$ linked to the level of quantum noise reduction by means of Eq. (2).

As noted in the discussion below Eq. (1), correlations between the SN and QBAN can alter the spectrum of the light noise in a manner that mimics a probe system with a downshifted resonance frequency.

Invoking this technique is of particular interest for sensing in the audio band, as straightforward engineering of a quantum-limited probe system with a low resonance frequency is challenging due to thermal and technical noise sources. The virtual shift is also a crucial element of the broadband quantum-noise reduction scheme for GWD beyond the SQL presented in Ref. 23. The frequency response of the GWD is close to the free-mass susceptibility $\chi_I \propto -1/\Omega^2$. The idea of Ref. 23 is to engineer an effective spin oscillator with the same susceptibility, $\chi_S \propto 1/\Omega^2$, as for the GWD (except for an overall sign flip), which can be accomplished by the virtual frequency downshift of the spin oscillator.

To explain how this virtual shift arises, we start by noting that the light spectrum resulting from a measurement of a spin oscillator is modified when $\phi \neq 0, \pi/2$ due to the cross-correlations between SN and QBAN, as captured by Eq. (1). However, such a squeezing spectrum does not readily reveal the performance of the spin oscillator in the aforementioned applications. Instead, the squeezing spectrum (e.g., Fig. 2b) should be rescaled to force-noise normalization (e.g., Fig. 2c), which directly shows the sensitivity of the measurement to forces acting on the spin oscillator. The renormalization is performed according to the Fourier-frequency-dependent transfer function that maps a force acting on the oscillator into the output light (the procedure is detailed in the SI). An elucidating analytical description of the force-normalized spectra is achieved by changing to a new basis of *uncorrelated* SN and QBAN light quadratures (see SI), yielding the effective susceptibility of the spin oscillator (assuming $\gamma_S \ll \Omega_S$)

$$\tilde{\chi}_S^{-1}(\Omega) = \frac{\Omega_S^2 - \Omega^2 - i\gamma_S\Omega}{\Omega_S} + \Gamma_S \sin(2\phi). \tag{4}$$

The virtual spring softening arises from the term $\propto \Gamma_S$ in Eq. (4) and results in the effective oscillator frequency $\tilde{\Omega}_S = \Omega_S\sqrt{1 + \Gamma_S \sin(2\phi)/\Omega_S}$ defining the minimum point in the force-normalized spectrum[23]. Whenever $-\pi/2 < \phi \, \text{sign}(\Omega_S) < 0$, an effective frequency downshift is implemented.

We observe the frequency shift of the initial $|\Omega_S|/(2\pi) = 18$ kHz in the range $|\Delta\Omega_S|/(2\pi) = |\tilde{\Omega}_S - \Omega_S|/(2\pi) \lesssim 2.1$ kHz with its sign depending on the sign of the effective mass of the oscillator. The maximal $\Delta\Omega_S$ is obtained at the homodyne detection phase set to $\phi = \pm \text{sign}(\Omega_S)\pi/4$. The size of the shift matches well the extracted experimental parameters of the system, mainly meaning the readout rate $\Gamma_S$. The ideal regime for application to GWD noise evasion is when $\Gamma_S$ exceeds $\Omega_S$, as it opens up the possibility to reduce the effective resonance frequency

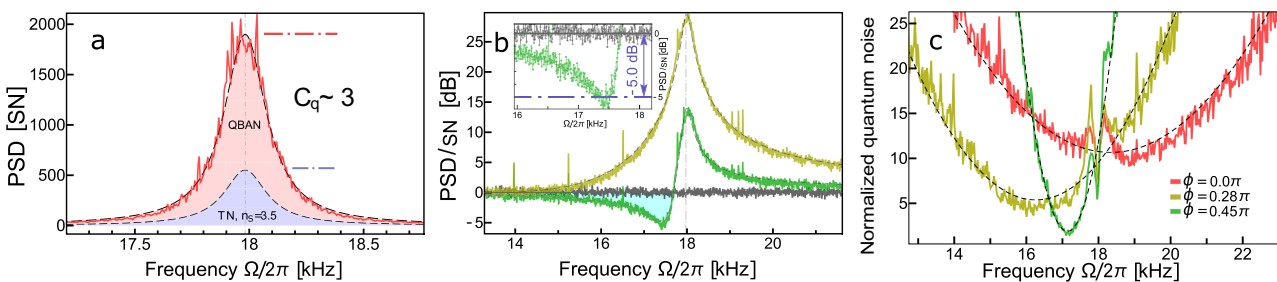

**Fig. 2 | Spin noise spectra at Larmor frequency $|\Omega_S|/(2\pi) = 18$ kHz. a** The homodyne phase is set to $\phi = 0$, corresponding to the detection of the phase quadrature of probe light (red curve). The fitting of experimental traces using noise model Eq. (1) is described in the text. Reconstructed quantum back-action noise (QBAN) and thermal noise (TN, defined by thermal occupation $n_S = 3.5$) are shown as the light red shaded area and the light blue shaded area, respectively. The ratio between QBAN and TN results in the quantum cooperativity $C_q = 3$. **b** The homodyne phase is adjusted to produce maximum ponderomotive squeezing (green curve) $S_S \lesssim -5$ dB (also shown in the inset) below the shot noise level (black curve). The yellow curve shows the spin noise at $\phi \approx -0.25\pi$ detection quadrature. Axes normalized to the shot noise of light [SN], represented in linear or decibel scale.

**c** Total force-normalized quantum noise of light (SN and QBAN) exhibiting the virtual tuning of effective resonance frequency $\tilde{\Omega}_S$, whose absolute value corresponds to the position of the minimum for each curve. The shift depends on the homodyne detection phase $\phi$, see Eq. (4), and is accompanied by a decreased effective readout rate $\tilde{\Gamma}_S = \Gamma_S \cos^2\phi$. In particular, the choice $\phi \approx -0.25\pi$ provides $\Delta\Omega_{S,1}/(2\pi) \approx -2.1$ kHz, whereas observation of maximized ponderomotive squeezing ($\phi_{opt} \approx -0.45\pi$) yields the smaller downshift $\Delta\Omega_{S,2}/(2\pi) \approx -1.2$ kHz. Apart from that, such force-normalized quantum noise leads to a decrease of the vertical offset (better sensitivity to an external signal) together with an increase of the steepness (reduced quantum-enhanced bandwidth) [see SI for details].

down to zero, $\tilde{\Omega}_S = 0$, which occurs at $\Gamma_S \sin(2\phi) = -\Omega_S$. Based on the present demonstration, we can envision a realistic spin oscillator with bare frequency $\Omega_S/(2\pi)$ in the kHz range whose susceptibility is modified by the virtual frequency shift so as to match the susceptibility of a free mass, characteristic of the GWDs.

### Quantum spin oscillator in the low-frequency acoustic range. Suppression of the near-DC noise

Having investigated the atomic spin oscillator in the upper audioband, we now target the lower acoustic range down to sub-kHz range. We find that a straightforward reduction of Larmor frequency down toward DC-frequencies by reducing the external magnetic field is accompanied by drastic reduction of ponderomotive squeezing that entirely disappears at $|\Omega|/(2\pi) \sim 10$ kHz. If the model Eq. (1) is used, the compromised performance of the spin oscillator can be accounted for by a boost of the thermal occupation $n_S$, consequently affecting $S_{TN}$ and reducing quantum cooperativity $C_q$. Searching for an explanation from a physical point of view, we envision that the incompleteness of the spin noise model Eq. (1) is due to the deviation of the light-spin interaction from the QND Hamiltonian $\propto a_1 \hat{S}_z \hat{j}_z$ in the near-DC frequency range. The description of the ground-state multiplet $F = 4$ of Cesium atoms requires extension beyond the two-level (spin-1/2) model[35] implied by the QND Hamiltonian. Such expansion involves alignment operators $\hat{j}_x^2 - \hat{j}_y^2$, $\{\hat{j}_x, \hat{j}_y\} \equiv \hat{j}_x \hat{j}_y + \hat{j}_y \hat{j}_x$ that couple to a probe field through the atomic tensor component proportional to the tensor polarizability $a_2$[36]. Accordingly, the following amendment to the QND interaction Hamiltonian must be included

$$\hat{H}_{int}^{(2)} \propto a_2 \left[ \hat{S}_y \{\hat{j}_x \hat{j}_y\} + \hat{S}_x \left( \hat{j}_x^2 - \hat{j}_y^2 \right) \right]. \tag{5}$$

The effect of the first term in the square brackets is centered around the Larmor frequency $\Omega_S$ and can be adjusted by the input polarization of light. The second term affects the spin noise at $\Omega = 0$ and $\Omega = 2\Omega_S$ since the matrix element $\left\langle F, m_{F,f} \left| \hat{j}_x^2 - \hat{j}_y^2 \right| F, m_{F,i} \right\rangle$ is non-zero for $|m_{F,f} - m_{F,i}| = 0, 2$, respectively[37]. We observe both the $\Omega = 2\Omega_S$ and $\Omega = 0$ spectral components (see Methods, Sec. *Spin alignment noise*)[38], but mainly focus on the latter, which we will refer to as 'DC noise'. The zero-frequency component amplified by the intensity noise of the probe laser spans up to $|\Omega|/(2\pi) \lesssim 10 - 20$ kHz, as shown in Fig. 3a. Consequently, the contribution of the DC noise to the noise budget leads to deterioration of the ponderomotive squeezing in the low audio-frequency band.

Crucially, we find that such DC noise can be strongly suppressed by minimizing the alignment term in the Hamiltonian, Eq. (5). In particular, one can increase the optical detuning $\Delta$ and benefit from the fast decline of $a_2$[36] which defines the strength of the alignment noise (see Fig. 3a). However, it should be taken into account that QBAN and thermal noise also depend on the detuning (Methods, Sec. *Spin noise spectra with effective masses*). Analyzing each term as a function of $\Delta$ (shown on Fig. 3b), we predict the existence of an optimal detuning $\Delta_{opt}$ yielding the best ponderomotive squeezing (see Methods, Sec. *Spin alignment noise* for details). We confirm it experimentally for the spin oscillator with the resonance frequency $|\Omega_S|/(2\pi) = 3$ kHz (see Fig. 3c). For such oscillator the increase of the detuning from an initial $\Delta_{in}/(2\pi) = 1.6$ GHz up to $\Delta_{opt}/(2\pi) \in 3.0 - 3.5$ GHz has resulted in the maximal level of ponderomotive squeezing $S_S(\Delta_{opt}) \lesssim -3$ dB (Fig. 4). A similar optimization of $\Delta$ for even lower Larmor frequencies resulted in $S_S = -2$ dB and $S_S = -1.3$ dB of quantum noise suppression below shot noise level at $|\Omega_S|/(2\pi) = 2$ kHz and $|\Omega_S|/(2\pi) = 1$ kHz respectively, shown on the lower panels of Fig. 4. The contribution of QBAN to the dynamics of the spin oscillator remains substantial down to the lowest acoustic frequency, although being reduced, as quantified by the extrapolated $C_q$ indicated in Fig. 4 (top panel).

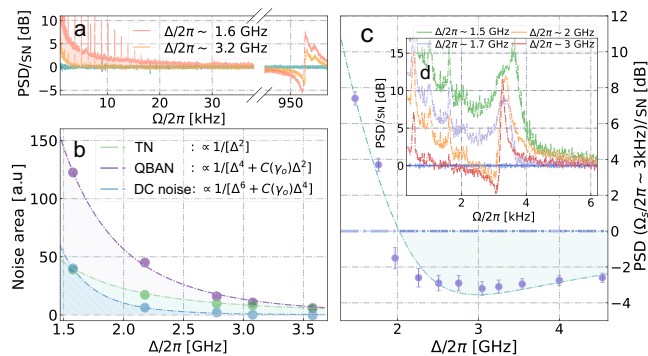

**Fig. 3 | Various contributions to the total spin noise budgets. a** Spectra of the light probing the spin ensemble reveal the strong near-DC component ($\Omega/(2\pi) \lesssim 20$ kHz, being clearly separated from the response at $\Omega_S$ (set to 1 MHz), leading to the reduction of ponderomotive squeezing in the acoustic frequency range. The DC-noise contribution decreases as the optical detuning $\Delta$ is increased. **b** Comparison of thermal noise (TN), quantum back action noise (QBAN), and DC-noise areas as a function of $\Delta$. **c** Influence of probe detuning $\Delta$ on the degree of ponderomotive squeezing measured at $|\Omega_S|/(2\pi) \approx 3$ kHz, where DC noise has a significant contribution to the noise budget. At the detuning optimal for ($\Delta_{opt}/(2\pi) \in 3.0-3.5$ GHz) the ratio between QBAN and uncorrelated noise sources (including DC noise) is maximized and the best squeezing $S_S \approx -3$ dB is observed. The error bars represent the uncertainty of extracted ponderomotive squeezing at specific detunning values. **d** Spin noise spectra at different detuning $\Delta$ in the optimal for ponderomotive squeezing detection phase $\phi_{opt}$.

## Discussion

We have experimentally demonstrated a macroscopic quantum spin oscillator in the acoustic frequency range. Quantum-backaction-dominated performance has been achieved for the oscillator with a negative effective mass. We have shown effective spring softening, an effect critical for the implementation of broadband quantum noise reduction in the acoustic and near-DC frequency bands relevant for various applications including gravitational wave detection beyond the SQL. We have identified the deleterious effect of the tensor spin polarizability on the low-frequency spin quantum noise and have found a way to minimize it by an optimal choice of detuning $\Delta$ of the probe light.

The reported results constitute an important milestone toward the implementation of the proposal[22,23] for suppression of the quantum noise in interferometer-type GWDs using a negative-mass atomic oscillator as a reference. Combining the spin oscillator at $|\Omega_S|/(2\pi) \lesssim 2$ kHz dominated by QBA with an effective downshift of the Larmor frequency $|\Delta\Omega_S|/(2\pi) \gtrsim 2$ kHz demonstrated in the upper audioband, we expect to emulate the motion of a free-mass object, operating the negative-mass spin oscillator with $\tilde{\Omega}_S$ approaching zero. Figure 5 illustrates the expected broadband noise reduction in the GWD signal below the SQL obtained by combining the spin ensemble and the entangled light source demonstrated in Ref. 24. The dark red curve presents the case of $C_q = 40$, corresponding approximately to the ratio $\Gamma_S/\gamma_{S,pb}$ in the present experiment, while assuming the absence of thermal noise $n_S = 0$, suppressed tensor noise, negligible optical losses and the power-broadening-dominant regime ($\gamma_{S,0} \ll \gamma_{S,pb}$). The effect of a moderate thermal noise $n_S = 3$, which reduces $C_q$ and adds extra uncorrelated noise, is shown by the light red curve. The orange dashed vertical line indicates the initial resonance frequency of the spin oscillator $|\Omega_{S,GWD}|/(2\pi) \approx 48$ Hz which is optimal for the implementation of the virtual frequency shift in the presented frequency range. The reduction of the intrinsic atomic linewidth $\gamma_{S,0}$ together with the mitigation of DC noise will make it possible to reach a sensitivity improvement of GWDs comparable to the predicted performance of other quantum-noise-evasion protocols[39]. The advantages of our approach in comparison to, e.g., achieving frequency-dependent

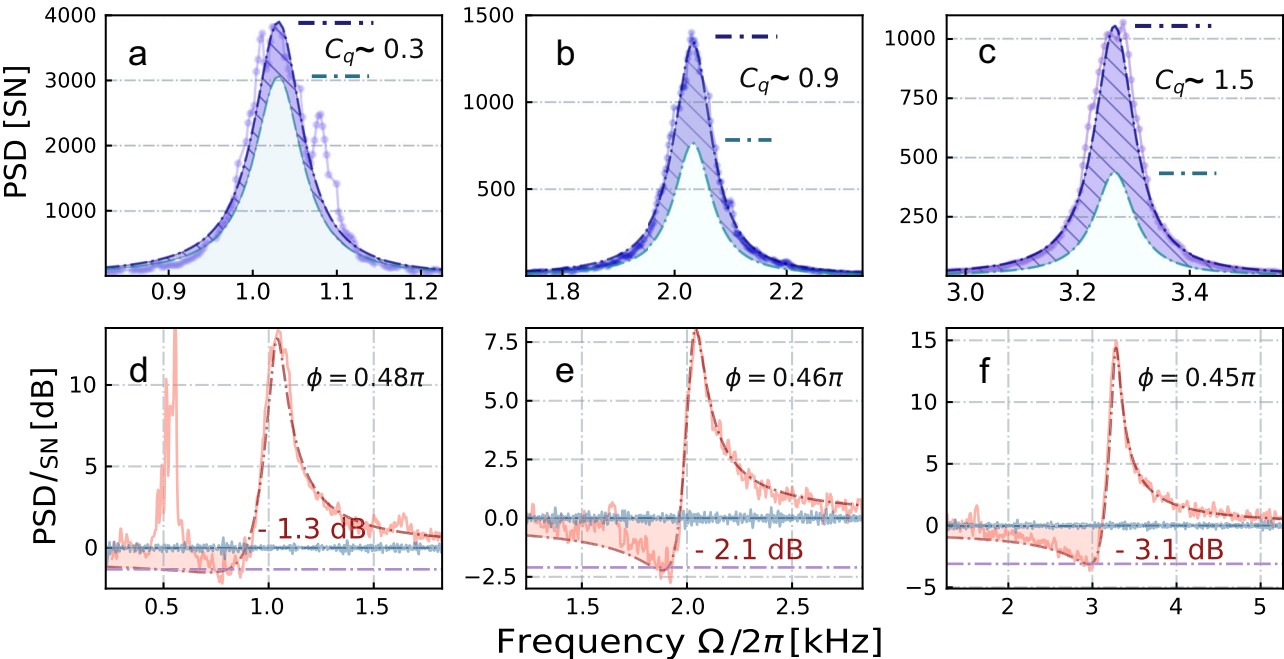

**Fig. 4 | Spin noise spectra recorded at lower audio sideband frequencies.** The spectra of the phase quadrature ($\phi = 0$) are displayed in the top panel, where the reconstructed quantum back-action noise (QBAN) and thermal noise (TN) are represented by the purple dashed area and the light green shaded area, respectively (**a–c**). The bottom panel, consisting of sub-figures (**d–f**), displays the case when the homodyne phase is adjusted to produce the strongest squeezing induced by the atomic ensemble. The level of ponderomotive squeezing is optimized by adjusting the optical detuning for each Larmor frequency, being gradually increased from $\Delta/(2\pi) = 3$ GHz for $|\Omega_S|/(2\pi) = 3$ kHz up to $\Delta/(2\pi) = 4$ GHz for $|\Omega_S|/(2\pi) = 1$ kHz. See comments in the text. Axes normalized to the shot noise of light [SN], represented in either a linear or decibel scale.

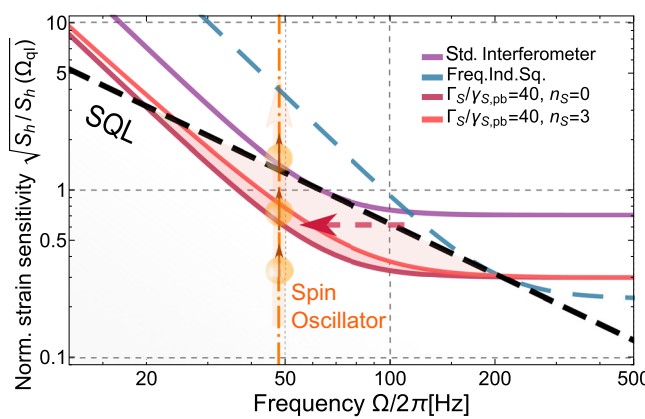

**Fig. 5 | The strain-referenced quantum noise $S_h$ of GWD with characteristic interferometer coupling rate $\Omega_{ql}/(2\pi) = 63$ Hz[18,23].** The sensitivity of a standard quantum-noise-limited interferometer (magenta curve) and the configuration with an injected frequency-independent 10 dB phase-squeezed vacuum state of light (blue dashed curve) is compared with the *projected* sensitivity of a joint measurement in the reference frame of a negative-mass spin oscillator linked to the GWDs by utilizing an entangled state of light (10 dB two-mode-squeezed vacuum state). Results for two different parameter configurations of the joint system are represented by the dark and light red curves, respectively. In both cases the Standard Quantum Limit (SQL) is expected to be surpassed, as indicated by the dashed black curve and highlighted by the shaded red areas. Further details are discussed in the text.

squeezing by means of a long filter cavity[40,41], include the tunability of the quantum noise evasion (via $\Gamma_S$, $\Omega_S$ and $\phi$) and its small physical footprint. Another possible advantage is the reduced effect of optical losses in the GWDs, which is due to the fact that only one of the two entangled modes propagates in the GWD, whereas the other mode interacts with the relatively low-loss spin ensemble[22,23].

In a broader perspective, the reported results are relevant for quantum sensing of particle mobility[42] or magnetic fields[43] in the acoustic range of sideband frequencies. The squeezed light source in the acoustic frequency range reported here has certain advantages compared to more traditional sources based on nonlinear optics[44]. It does not require powerful lasers and nonlinear crystals and is characterized by intrinsic phase stability due to collinear propagation of the coherent carrier and quantum fluctuations. The robust and tunable squeezed light source reported here is relevant for quantum magnetometry[45], especially for biomedical applications where signals in the sub-kHz range often prevail[26]. In the field of hybrid optomechanics, coupling of the atomic spin oscillator to a trapped dielectric nano-particle would allow the optical backaction-evading measurement of mechanical forces in the ~ 1 – 200 kHz frequency range[46].

## Methods

### Atomic vapor cell and PCB coils

The spin ensemble, consisting of approximately $N_S \approx 10^{10}$ ~ $10^{11}$ Cesium-133 atoms, is contained in an antirelaxation-coated (C30+) rectangular channel ($2 \times 2 \times 80$ mm³) providing a good balance between large quantum cooperativity $C_q$[31] and maintaining low-frequency quantum-noise-dominated performance for our experiment. The spin-preserving coating grants a room temperature dark decoherence rate of ~ 50 Hz during the experiment and the connection to a Cesium atom reservoir allows adjusting the vapor density $\rho$ based on the operational temperature. The vapor cell is placed in magnetic fields provided by specially designed PCB coils. The inner bias magnetic field is generated by a coil system that combines a concave and convex parabolic magnetic fields with a linear-gradient field. This system is driven by an ultra-low-AC-noise current source, achieving an inhomogeneity of <0.1‰ within the cell volume[47] (refer to Supplementary Note 1 for more details). The setup is positioned in a 5-layer magnetic shield protecting the spins from perturbations from the external DC and RF magnetic fields. The setup with freely adjustable PCB coils

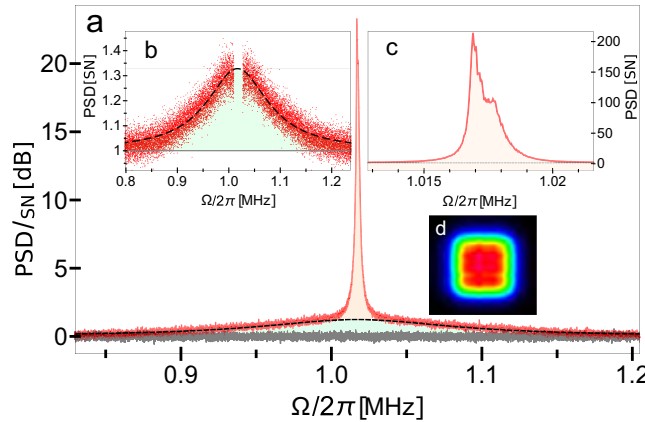

**Fig. 6 | Observation of atomic spin noise at $|\Omega_S|/(2\pi) \approx 1$ MHz. a** PSD of spin noise including zoomed broadband noise (**b**) and narrowband response (**c**) represented with light green and orange areas, respectively. Vertical axes are normalized to the shot noise [SN] of light. **d** The probe beam has a $1.65 \times 1.65$ mm$^2$ square top-hat beam profile in order to reduce the broadband noise contribution.

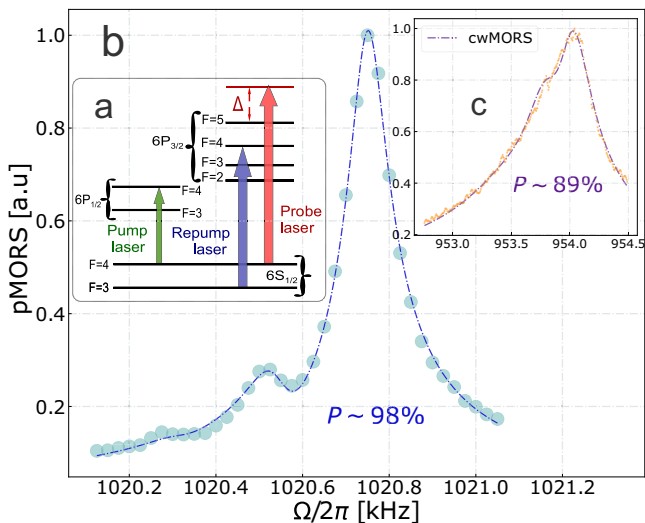

**Fig. 7 | Preparation and characterization of the collective spin oscillator. a** The structure of atomic levels, providing an overview of optical pumping. **b** and **c** The spin polarization $\mathcal{P}$ extracted from magneto-optical resonance spectroscopy (MORS) method in the pulsed and continuous regimes, respectively. The MORS signals are shown in arbitrary units (a.u.).

system allows tuning the Larmor frequency from a few Hz up to 1 MHz without obviously affecting the intrinsic line-width $\gamma_{S,0}/(2\pi)$.

## Broadband noise reduction (BNR)

When a linearly polarized probe light interacts with a spin ensemble and records the dynamic of the collective spin system, the measured spin noise spectrum (SNS) in Fig. 6a would be affected by various dephasing mechanisms, such as wall collision, the probe beam size, and atomic motion diffusion characteristics[31,48–50]. Therefore, the SNS from an atomic vapor cell is a combination of Lorentzians $(S_{\text{total}}(\Omega) \rightarrow \sum \Gamma_i \chi_i(\gamma_{Si}, \Omega))$ with the individual weights $(\Gamma_i, \gamma_i)$ which correspond to the overlap of the probe beam spatial profile (e.g., a Gaussian mode) with each of the spin diffusion modes. The pronounced narrowband noise spectrum in Fig. 6c (orange area) originated from the sum of the slowly decaying modes, and the broadband spin response floor Fig. 6b (green area) is due to the modes which decay rapidly due to the motion of atoms in and out of the probe beam during the measurement. With the help of a diffractive beam shaper and a telescope system, we could produce a $1.65 \times 1.65$ mm$^2$ square top-hat beam (as shown in Fig. 6d) collimated along 8 cm (corresponds to a cell filling factor of 72%). The increased filling factor for the rectangular cell channel helps to reduce the broadband noise down to <0.3 in shot noise units [SN] and to improve the relative amplitude ratio between the narrowband and broadband response up to ~600, making the contribution of the $S_{bb}$ term in Eq. (1) negligible.

## Preparation and characterization of atomic state

The Hamiltonian for an ensemble of atomic spins with a collective angular momentum $\hat{\boldsymbol{J}} = \sum_{k=1}^{N} \hat{\boldsymbol{j}}^k$ in the external magnetic field $\boldsymbol{B}$ is $\hat{H}_B \sim -\boldsymbol{J} \cdot \boldsymbol{B}$. All $N$ atoms are initially prepared in the state $6S_{1/2}$, $|F = 4, m_F = -4\rangle$ or $|F = 4, m_F = 4\rangle$, where $m_F$ denotes the Zeeman sublevel within the hyperfine manifold $F$. The ensemble is then polarized along the $x$-axis, so that the component $\hat{J}_x$ becomes a macroscopic variable $\hat{J}_x \rightarrow J_x = \hbar F N/2$. Within the Holstein-Primakoff approximation, the spin precesses in the $yz$-plane $\sim \Omega_S(\hat{J}_z^2 + \hat{J}_y^2)$ at the Larmor frequency $\Omega_S \sim |\boldsymbol{B}|$. The collective spin can be co-oriented $(J_x > 0)$ or counter-oriented $(J_x < 0)$ with respect to $\boldsymbol{B}$. This leads to opposite directions of rotation of the $\hat{J}_{y(z)}$-components, or equivalently, to the opposite signs of $\Omega_S$. This situation is commonly referred to a spin oscillator with a negative or positive effective mass[17].

The detailed configuration of atomic levels without Zeeman splitting is depicted in Fig. 7a which outlines the pumping scheme. Circularly polarized pump and repump lasers are tuned to the $|6S_{1/2}, F = 4\rangle \leftrightarrow |6P_{1/2}, F' = 4\rangle$ and $|6S_{1/2}, F = 3\rangle \leftrightarrow |6P_{3/2}, F' = 4\rangle$ transitions respectively as in Fig. 7, which corresponds to the D1 and D2 lines. The applied method of atomic polarization characterization is based on magneto-optical resonance spectroscopy (MORS,[34]). The spacing between adjacent Zeeman sublevels on the ground hyperfine level follows the equation

$$\frac{E_{F,m+1} - E_{F,m}}{\hbar} = \Omega_S + \Omega_{QZS}(2m+1), \qquad (6)$$

where $\Omega_{QZS} \sim \Omega_S^2$ refers to Quadratic Zeeman splitting effect. Consequently, Zeeman resonances can be resolved provided their small linewidth compared to $\Omega_{QZS}$. This condition turns out to be fulfilled if the bias magnetic field is boosted and the resonance frequency $\Omega_S/(2\pi)$ is set to MHz range ($|\boldsymbol{B}| \sim 3$ G). The Zeeman transitions are excited by applying an AC-magnetic field, the resulting spin response is recorded onto the probing optical field and is then read out by means of balanced polarimetry. The strength of the transitions between Zeeman sublevels depends on their populations. Therefore, the orientation of the spin ensemble, as quantified by the spin polarization $\mathcal{P}$, can be characterized using the MORS signal.

We extract the spin polarization $\mathcal{P} \approx 98\%$ in Fig. 7b (equivalent to the thermal occupation of $n_S \sim 0.15$) using the pulsed MORS with 1mW of probe light. The orientation goes down to $\mathcal{P} = 89\%$ (Fig. 7c, yielding $n_S \sim 0.6$) in the regime of continuous probing under the same optical power[31]. The repump power $P_{re} \approx 5$mW was conditioned upon the maximum available laser power, whereas the pump power $P_p \approx 50\,\mu$W was chosen after the optimization of ponderomotive squeezing at $|\Omega_S|/(2\pi) \sim 1$ MHz. The power broadening from the pump and repump lasers contribute <100 Hz decoherence to the spin linewidth. From the pulsed MORS we estimate the intrinsic linewidth $\gamma_{S0}/(2\pi) \approx 150$ Hz which contains all decay contributions except for the power broadening induced by the probe field. Using Eq. (1), we fit the spectra of light probing the spin ensemble at

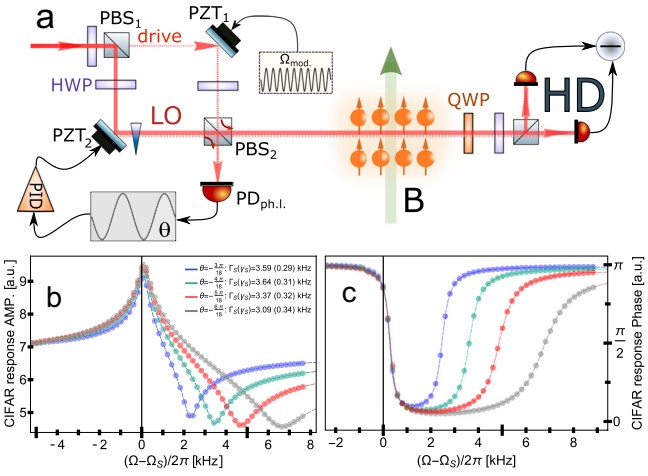

**Fig. 8 | Implementation of coherently induced Faraday rotation (CIFAR) technique. a** The layout of the experimental setup. **b** and **c** Amplitude and phase responses at different modulation phases $\theta$. The extracted readout rate $\Gamma_s$ along with the total decay rate $\gamma_s$ are indicated on the figure legends. The signal is shown in arbitrary units (a.u.).

Larmor frequency $|\Omega_S|/(2\pi) = 18$ kHz. The retrieved thermal occupation $n_S = 3.5$ is larger than the result obtained from the calibration by MORS in the continuous regime. The $n_S$ extracted from the full spin model might then be treated as an effective thermal occupancy that includes additional noise sources not accounted for in Eq. (1), for example, intensity noise of the probe laser.

## Calibration of readout rate

To calibrate the spin measurement rate $\Gamma_S$ and damping rate $\gamma_S$, we investigate the atomic response to strong modulation of the probe light polarization. The outlined technique is referred to as Coherently induced Faraday rotation (CIFAR)[33]. The experimental setup is shown in Fig. 8a. A weak linearly polarized optical field denoted as 'drive' is phase-modulated at frequency $\Omega_{mod}$ using the piezoelectric transducer $PZT_1$ and subsequently overlapped with the orthogonally polarized Local Oscillator (LO) on a polarizing beamsplitter $PBS_2$. One of the output modes of $PBS_2$ thus contains the field $\hat{Q}_{L,in}^{mod}(\theta) \sim (\hat{X}_{L,in}\sin\theta + \hat{P}_{L,in}\cos\theta)\sin(\Omega_{mod}t)$ with an arbitrary modulated polarization quadrature. The phase angle $\theta$ is set by the phase lock loop between LO and drive fields with a feedback signal applied to the piezo element $PZT_2$ in one of the interferometer arms. The optical field $\hat{Q}_{L,in}^{mod}(\theta)$ probes the atomic oscillator and is then detected with the same balanced polarimetry detection setup. Scanning the modulation frequency $\Omega_{mod}$ around the Larmor frequency $\Omega_S$, one obtains the characteristic shape of the measured spectrum signal $S_{CIFAR}(\Omega_{mod})$ (shown in Fig. 8b, c) that provides information about $\Gamma_S$ and $\gamma_S$. However, the correctness of the extracted parameters strongly depends on the precise knowledge of the modulation phase $\theta$. To account for that, we perform the fit of $S_{CIFAR}(\Omega_{mod})$ at several points of locked $\theta$ and obtain results for $\Gamma_S$ and $\gamma_S$ as shown on the Fig. 8b, c. The uncertainty ~15% on both parameters is mainly attributed to imperfect calibration of $\theta$, which is limited by the software of an FPGA board in the phase lock loop. This circumstance might also address the discrepancy between the values of the readout rate from CIFAR technique and from the fit of the full spin noise model Eq. (1). Therefore, we consider the CIFAR calibration as a rough estimation of measurement and damping rates and use them as initial parameters for the full spin noise model.

## Spin noise spectra for effective positive and negative masses

The main goal of a measurement reported in this section is to reveal the difference between a spin oscillator with an effective negative and positive mass. We operate the atomic ensemble in the Zeeman resolved regime (magnetic field is set to $|\boldsymbol{B}| \approx 3$G, giving $|\Omega_S|/(2\pi) \approx 1$ MHz, as for MORS calibration, Sec. C) and study spin noise spectra (presented on the Fig. 9c, d), when the system is driven by quantum-noise-limited light without an applied AC-magnetic field. It is then possible to see the consequences of finite spin polarization, and hence the populations of Zeeman sublevels different from $|m_F = 4\rangle$ in the $F = 4$ hyperfine multiplet. Specifically, we observe several peaks around $|\Omega|/(2\pi) \approx 960$ kHz. Using Eq. (6), we identify two peaks centered at $\Omega_{S1a}$ ($\Omega_{S1d}$) and $\Omega_{S1b}$ ($\Omega_{S1c}$) as the transitions $|m_F = -4\rangle \leftrightarrow |m_F = -3\rangle$ ($|m_F = 4\rangle \leftrightarrow |m_F = 3\rangle$) and $|m_F = -3\rangle \leftrightarrow |m_F = -2\rangle$ ($|m_F = 3\rangle \leftrightarrow |m_F = 2\rangle$) respectively within the $F = 4$ hyperfine multiplet. The prevailing $|F = 4, m_F = 4\rangle \leftrightarrow |F = 4, m_F = 3\rangle$ transition (Fig. 9d) corresponds to the inverted spin population since the majority of atoms occupy $|m_F = +4\rangle$. Thus, the negative-mass oscillator[6] is revealed. Whereas the strong $|F = 4, m_F = -4\rangle \leftrightarrow |F = 4, m_F = -3\rangle$ transition (Fig. 9c) corresponds to the positive-mass oscillator.

Moreover, using the spin oscillator at $|\Omega_S|/(2\pi) \approx 1$ MHz, we extract QBAN and thermal noise (TN) by calculating their integrated areas and subsequently calibrate them as function of the optical detuning $\Delta$. From the model Eq. (1), one can infer $\int_\Omega S_{TN}d\Omega \sim \gamma_S\Gamma_S \int |\chi_S(\Omega)|^2 d\Omega = \Gamma_S \sim A/\Delta^2$ and $\int_\Omega S_{QBAN}d\Omega \sim \Gamma_S^2 \int |\chi_S(\Omega)|^2 d\Omega = \Gamma_S^2/\gamma_S \sim A^2/[\Delta^2(\gamma_{S,0}\Delta^2 + C)]$ respectively. Here $\Gamma_S = A/\Delta^2$, $\gamma_S = \gamma_{S,0} + C/\Delta^2$, where $A$, $C$ and $\gamma_{S,0}$ are constant parameters independent of $\Delta$ as well as the vector polarizability $a_1 \approx 1$ in the explored range of detunings. We validate the expected behavior both for $\int_\Omega S_{QBAN}d\Omega$ and $\int_\Omega S_{TN}d\Omega$ while varying $\Delta$, as shown in Fig. 3b.

## Spin alignment noise

An atomic spin ensemble driven by Hamiltonian Eq. (5) demonstrates the distinctive features of linear birefringence. At the quantum level, the composite dynamics of the spin alignment interaction causes several phenomena, such as a tensor-induced Stark shift of the oscillator's Larmor frequency, cooling or amplification of the spin state, and even spin dynamics beyond the oscillation frequency. In this section, we will give an overview of the influence of each alignment operator on the atomic spin dynamics.

We start with the term $\{\hat{j}_x\hat{j}_y\}$. After applying the approximation $\{\hat{j}_x\hat{j}_{y(z)}\} \approx 7\hat{j}_{y(z)}$ valid in a two-level model, the total interaction is described by[31]

$$\hat{H}_{int} \propto a_1\left(\hat{S}_z\hat{J}_z + \mathcal{E}_S\hat{S}_\perp\hat{J}_y\right),$$
$$\mathcal{E}_S = -14\left(\frac{a_2}{a_1}\right)\cos(2\alpha), \tag{7}$$

where Stokes operators were redefined as $[\hat{S}_\parallel, \hat{S}_\perp]^T = \boldsymbol{R}(2\alpha)[\hat{S}_x, \hat{S}_y]^T$, where $\boldsymbol{R}(2\alpha)$ is the rotation matrix. The presence of the $\hat{S}_\perp\hat{J}_y$ term added to the Faraday rotation $\hat{S}_z\hat{J}_z$ means that the interaction deviates from the QND interaction. It affects the response of the atomic system recorded onto the phase light quadrature $\hat{P}_{L,out}$ (see Fig. 10a). Such impact might be seen as an effective change of the QND readout rate $\Gamma_S$ and inducing a dynamic contribution to the damping rate $\gamma'_S/2 \sim \gamma_S/2 + \mathcal{E}_S\Gamma_S$. Consequently, the maximal level of ponderomotive squeezing is altered (Fig. 10b), when $\hat{Q}_L(\phi_{opt})$ is selected. Finally, the amplitude output light quadrature $\hat{X}_{L,out}$, being a QND variable otherwise, is now also disturbed. This is manifested in a characteristic dip/peak as demonstrated in Fig. 10c. At the same time, we notice that the strength of the $\hat{S}_\perp\hat{J}_y$ term is controlled by the angle $\alpha$ of the probe input polarization. In the present experiment we wish to work at the QND configuration, which is set by rotating a half-wave plate in front of

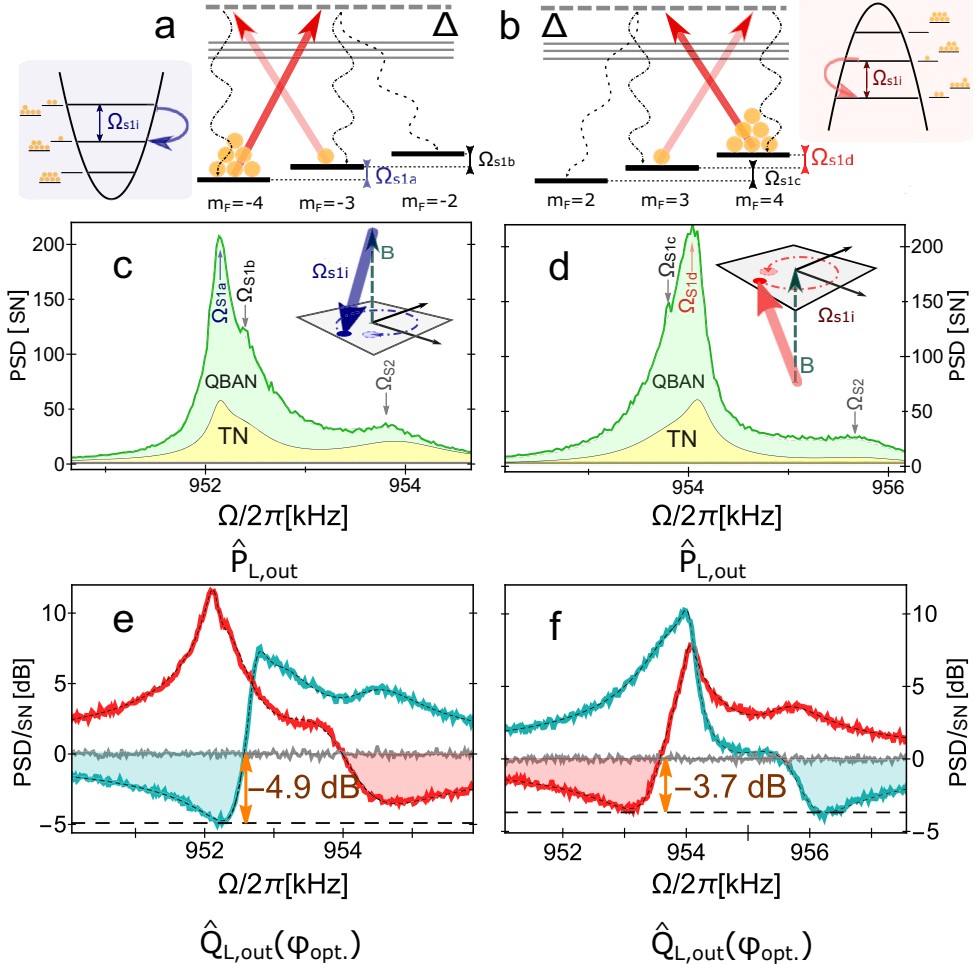

**Fig. 9 | The atomic oscillators with positive and negative effective masses.** The left (**a**, **c**, **e**) and right (**b**, **d**, **f**) columns display the configurations of the spin system with positive and negative mass, respectively. **a, b** Atomic ensemble is described as a harmonic oscillator within the 2-level-system approximation (either of $m_F = \pm 4$ and adjacent Zeeman sublevels on hyperfine level $F = 4$). If a single excitation lowers the energy of the system (**b**), then the oscillator has an effective negative frequency (mass). **c, d** The spectra of the optical field after probing the atomic spin oscillator ($|\Omega_S|/(2\pi) = 0.96$ MHz), of which the phase quadrature is detected ($\phi = 0$). We distinguish the positive- (**c**) and negative-mass (**d**) configurations, comparing the frequency of the strongest transition $\Omega_{S1a}$ ($\Omega_{S1d}$) to the other transitions from the $F = 4$ multiplet [only $\Omega_{S1b}$ ($\Omega_{S1c}$) can be identified]. In addition, we also observe a third peak

in the spectra, centered at frequency $\Omega_{S2}$ that is always higher than $\Omega_{S1i}$, regardless of the sign of the mass of the oscillator at $F = 4$. This component presumably arises due to inhomogeneity of magnetic field across the atomic cell and represents unresolved Zeeman structure. Insets: the sign of the resonance frequency defines the orientation of rotation in phase space. **e, f** Adjustment of homodyne detection phase $\phi = \phi_{\text{opt}}$ allows for the observation of ponderomotive squeezing. Green and red curves correspond to the choices $\phi = -|\phi_{opt}|$ and $\phi = +|\phi_{opt}|$ respectively and compared to SN level (gray trace). The negative-mass oscillator displays reduced ponderomotive squeezing, allegedly due to an increased spin damping rate caused by extra magnetic inhomogeneity broadening with a sub-optimal current ratio for the magnetic coils. Axes are normalized to the shot noise [SN] of light.

the cell and making the spectrum of $\hat{X}_{L,\text{out}}$ flat around $\Omega_S$ corresponding to $\mathcal{E}_S \approx 0$ (also depicted in Fig. 10c). We note that the alignment operator studied here is also responsible for the tensor Stark shift effect moving the resonance frequency $\Omega_S$ (clearly seen in Fig. 10a). It has to be taken into account when estimating the size of the virtual frequency shift by cross-correlations between SN and QBAN.

We now analyze the term $\hat{j}_x^2 - \hat{j}_y^2$. It is conceivable that the spin ensemble can sense fluctuations of the probe laser via a mechanism responsible for the tensor interaction ($\sim a_2$)[51]. In particular, coupling through the alignment operator $\hat{j}_x^2 - \hat{j}_y^2$ explains the abrupt rise of noise centered at zero frequency, being clearly separated from the Larmor peak, as shown in Fig. 10d ($|\Omega_S|/(2\pi) = 43$ kHz). However, the DC noise component has a tangible overlap with the Larmor peak shifted down to the acoustic range. In this case the QBAN dominated dynamics and the ponderomotive squeezing are compromised.

We study the detrimental influence of DC noise on the ponderomotive squeezing and introduce the term $S_{DC}$ that should be included in the spin model Eq. (1). We then explore $S_{DC}$ as a function of the

detuning $\Delta$ in the manner it was done for QBAN and thermal noise. Having in mind the tensor interaction, we expect $S_{DC} \sim (a_2/a_1)^2 (\Gamma_S)^2 |\chi_{S,DC}(\Omega)|^2$, where $\chi_{S,DC}(\Omega)$ is the susceptibility function that defines the spectral shape of DC noise. We model $\chi_{S,DC}(\Omega)$ by a Lorentz peak with center frequency located at $\Omega = 0$. Consequently, one may surmise $\int |\chi_{S,DC}(\Omega)|^2 d\Omega \sim \int |\chi_S(\Omega)|^2 d\Omega$ if the mechanisms forming decay rate $\gamma_S$ are still valid for $S_{DC}$. Finally, we obtain the expression $\int_\Omega S_{DC} d\Omega \sim (a_2/a_1)^2 \int_\Omega S_{QBA} d\Omega \sim 1/\left[\Delta^4 \left(\gamma_{S,0}\Delta^2 + C\right)\right]$ for the integral area of DC noise, using the approximations $a_2 \sim 1/\Delta$ and $a_1 \sim 1$. Such dependence on the detuning is validated on the Fig. 3 for the experimental data. The next step is to exploit the approximation given by Eq. (2) for the optimized ponderomotive squeezing and add $S_{DC}$. This leads to the formula

$$S_S \approx 1 - \eta \frac{C_q(\Delta)}{C_q(\Delta) + 1} + \frac{D}{\Delta^r} \tag{8}$$

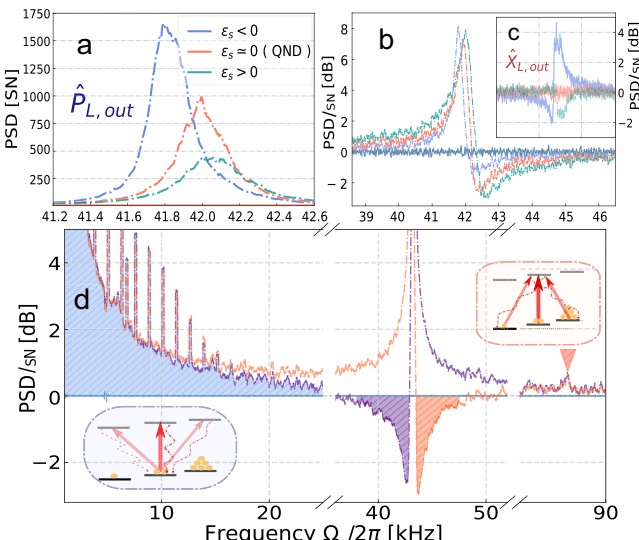

**Fig. 10 | Tensor alignment spin noise. a–c** The contribution of alignment noise $\propto \mathcal{E}_S$ (defined in Eq. (7)) manifested around the resonance frequency $|\Omega_S|/(2\pi) = 43$ Hz. **a** The measurement of the phase quadrature $\hat{p}_{L,\text{out}}$. **b** The measurement of $\hat{Q}_{L,\text{out}}(\phi_{\text{opt}})$, corresponding to the strongest ponderomotive squeezing. **c** The measurement of the amplitude quadrature $\hat{x}_{L,\text{out}}$. The coefficient $\mathcal{E}_S$ is varied by changing the input polarization angle $\alpha$. The QND interaction is set once the tensor contribution is minimized in the amplitude quadrature of light, as shown on (**c**). **d** Apart from the dispersive signal centered at $|\Omega_S|/(2\pi)$, the peak at twice Larmor frequency is visible together with noise enhancing toward $\Omega = 0$ (blue dashed area). The last two effects presumably originate from tensor interaction. Axes are normalized to the shot noise of light [SN].

where and $C_q \sim A/\left(C + \gamma_{S,0}\Delta^2\right)$ as was deduced in Sec. E. Note that we simplify the expression for DC noise and use $S_{DC} = D/\Delta^r$ ($r \in [4,6]$) in order to reduce the number of parameters in the model of the spin noise budget. The expression Eq. (8) states that there exists an optimal point $\Delta_{\text{opt}}$ which minimizes $S_S$. The $\Delta_{\text{opt}}$ is defined by the actual values of all coefficients in Eq. (8) and appears to be $\Delta_{\text{opt}}/(2\pi) \in 3 - 4$ GHz for a spin oscillator in low acoustic range and the chosen set of parameters (the example for $\Omega_S/(2\pi) = 3$ kHz is shown above). Exceeding this level brings us to the regime where reduction of the DC-noise term cannot compensate for the decline of $S_{QBA}/S_{TN}$ due to the significance of the intrinsic spin linewidth $\gamma_{S,0}$.

As a final remark, we note that the amount of DC noise depends on the phase of the detection quadrature. In particular, $S_{DC}$ is maximized in the amplitude Stokes quadrature, thus having a direct impact on the ponderomotive squeezing spectrum. In contrast, the DC noise is not present when the phase Stokes quadrature is observed. Also, it seems to be independent of the input light polarization (angle $\alpha$). Those effects require further investigation.

## Data availability

The data presented in the figures have been deposited in the University of Copenhagen repository under the link: erda.ku.dk/archives/7e6a369742d5d657d8db79967dec061a/published-archive.html.

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

## Acknowledgements

We gratefully acknowledge conversations with R. Thomas, M. Parniak, R. Schmieg, M. Zugenmaier, and J. Appel. We are grateful to Mikhail Balabas for his invaluable contribution in fabricating the alkene-coated vapor cell used in this experiment. We extend our appreciation to Ryan Yde for his contributions to the initial stages of the experiment. This work was funded by the European Research Council (ERC) under the Horizon 2020 (grant agreement No 787520) and by VILLUM FONDEN under a Villum Investigator Grant no. 25880. J.J. thanks the CSC for their support (201906140180). V.N. was partially supported by Russian Quantum Center.

## Author contributions

J.J., V.N., and T.B.B. performed the experiments. E.Z. contributed to the theory of virtual frequency shift (virtual rigidity), J.H.M. contributed to the experiment, E.S.P. led the project. All authors contributed to writing the paper. J.J. and V.N. contributed equally to this work.

## Competing interests

The authors declare no competing interests.
