## [Peer Review File · Nature Communications]

Acoustic frequency atomic spin oscillator in the quantum regimeREVIEWER COMMENTS

Reviewer #1 (Remarks to the Author):

Atomic spin oscillators are unique systems that allow to mimic mechanical oscillators, at the same time reaching unusual dynamics, such as that of a negative mass oscillator. Atomic spin system exhibit many of the conventional optomechanical effects, such as ponderomotive squeezing. This finds application in quantum optomechanical systems, where coupling a conventional oscillator to the atomic spin system allows to evade quantum back-action and thus significantly increase the sensitivity or bring mechanical oscillators in unusual quantum states. In [19], some of the authors proposed to cancel quantum back-action in gravitational-wave detectors, thus achieving an enhancement of the sensitivity to gravitational waves. This was proposed as an alternative to a conventional frequency-dependent squeezing generation using nonlinear crystals and large-scale optical filter cavities. The difficulty of achieving this with atomic spin ensemble was in matching the free test mass response in GW detectors. That required reaching a very low oscillation frequency with the spin ensemble, from typical 1MHz down to a few Hz, a very difficult task. In [19] it was proposed that measuring an optimal quadrature on the homodyne detector would introduce a so-called “virtual rigidity”, which would utilize quantum correlations between quantum radiation-pressure noise and shot noise to bring the effective frequency down into the required regime. This would still require bringing the oscillators to a lower frequency, but to a more feasible kHz domain. That is what the authors demonstrated experimentally in the present manuscript. Operating the atomic spin system at low frequencies in quantum regime was presented for the first time, in an impressive demonstration of the versatility of the setup, together with a detailed investigation of its various properties, specific to this regime.

The results are scientifically sound, presented with sufficient amount of detail and clarity. The methodology is adequate, and enough detail is provided for reproducing the work. I could not see any flaws in the data analysis, interpretation or conclusions.

I do, however, find the presentation to be somewhat lacking. I have two major comments, a several minor ones:

1. The authors motivate their work as relevant for gravitational-wave detectors. However, this motivation looks very focused on a specific narrow application, and even in this application it is far from being a competitor to the existing approaches. It is certainly a very interesting concept, but e.g. Ref. [19] does not present a clear reason for the approach to be practical for GWDs. The benefit there remains lower than the current approaches, and it also arguably lacks in versatility (e.g., as far as I'm aware, it's not applicable to the detuned interferometers). In my view, this weakens the motivation, especially when talking about the potential impact. I can imagine there is a broader impact on the field in general, but only very little space is dedicated to it at the very end of the text.

2. The overall clarity of the paper could be improved. Many things assume the readers to be familiar with other works by the authors. In particular, I'd like to point out a few things:

- The term "virtual rigidity" is never clearly explained. While people in the context of theory behind GW detection know what is meant, it is very unclear for the general audience. It's necessary to explain what is virtual about it, and why it is even called a rigidity. I would actually suggest to go away from this term altogether. While a nice analogy, it also is prone to misinterpretation: no real change in the system's dynamics is happening, there's "only" the effect appearing during the measurement.

- The central result presented by the authors is the quantum behavior of the system at audio band frequencies. However, the authors never explain (in the main text) how they actually tune the Larmor frequency down to 6kHz. It's implied by its proportionality to B , but it's never clarified. What is new about this frequency? Why was it not possible to go there before? What are the limitations (they are mentioned in the text, but not in a well structured way)?

- How does the virtual rigidity connect to the observed ponderomotive squeezing level? The authors observed 0.8dB of squeezing at 6kHz, what frequency shift would it translate to with virtual rigidity? Why wasn't this observed in the experiment?

- There's a lot of incoherence between the different parts of the paper. E.g. sometimes back-action noise is called QBA, sometimes QBAN, sometimes BAN. Spectral densities have different normalizations, which are not very well explained. What's the difference between PSD[dB] and PSD[SN]? Sometimes virtual rigidity is called virtual spring.

- The structure of the paper is not very clear. The authors announce the oscillators in

acoustic band, but make the first part about the MHz band (which were published elsewhere before). Virtual rigidity appears in the next section, and it's is again for MHz oscillators, while it would be most relevant for the kHz ones, if the paper tries to align it with proposal in [19]. Most of the rest of the paper is about observation of quantum features at audioband frequencies and technical issues associated with that, and it's not clear how it connected to observing virtual rigidity. I understand, that the intention is to say that virtual rigidity is demonstrated to be on the kHz level, and thus shifting the Larmor frequency down to kHz level would allow to make the effective frequency close to zero. This is hard to understand from text.

This all makes the paper quite hard to follow. I would recommend the authors to give a brief introduction to their main results already in the beginning of the paper, possibly extending the focus from GWDs to a broader picture (in the view of my point 1). Possibly, there exists a better structure for the paper (e.g. start with audioband observations to show that a low-frequency oscillator is possible, and then talk about what kind of virtual rigidity is required, and then demonstrate its existence for a high-frequency system).

Overall, I find the results to be quite beautiful. They deserve to be seen by a broad community and to make an impact on experiments and theoretical investigation. They would be appropriate for publication in a high-ranking journal, such as Nature Communications, but the current presentation needs to be improved.

Here're some minor comments:

1. On Fig.2 the squeezing is 4.9dB, in the text it's 4.8db.
2. In the same place it's stated that the squeezing for the negative mass is only slightly reduced. In practice it's more than 1dB difference, which is quite significant. Also, isn't it the squeezing for the negative mass that is relevant? Then it's somewhat misleading.
3. Fig.2 (e,f) is not color coded. Some legend would be useful.
4. Fig.3 lacks units on the y axis. Also it should be explained more clearly why another normalization is used (it's mentioned in the text, but not explained).
5. In Fig.5, the meaning of the red shaded area in (c) is not clear.

6. In Fig. 7, the meaning of the yellow circles & arrow is not clear (is it a representation of the spin system? May be then simply labeling it as Ω_s is sufficient).

Reviewer #2 (Remarks to the Author):

This work reports the study of noise spectrum of atomic spin oscillator in the acoustic band, and the ponderomotive squeezing and back-action reduction is demonstrated down to 6kHz. This is a notable technical achievement, as any experimentalist knows the Sisyphus nature of $1/f$ noise. The motivation and focus of current study seems to be mostly on potential applications for gravitational wave detection, although I think the paper would benefit from a more extensive discussion of other potential applications that may benefit from low-frequency quantum noise control. I find the presented results interesting and potentially impactful for quantum sensor community. However, I found the paper occasionally hard to follow due to some peculiarities of writing style and language. For example, I am not sure what is meant by the term "imprecision noise" used several times in the manuscript. Fig. 2 uses notation "BAN" and "TH" which I presume mean back-action noise and thermal noise, but in the text QBA abbreviation is used, and TH is not explained. Also, I must confess I had particularly hard time following the section "Frequency downshift of the spin oscillator by virtual rigidity". I am not an expert in optomechanics, but the heavy use of specialized terminology made it really hard for me to understand the relevance of this discussion to the rest of the manuscript. I think a more streamlined presentation with clearly formulated objectives would benefit readers who are, like me, may benefit from this work but not necessarily fully immersed into the details of the atomic spin oscillator performance.

Reviewer #3 (Remarks to the Author):

In this manuscript, Jia et al. report an experimental demonstration of an atomic ensemble in a vapor cell, which behaves as a spin oscillator in the presence of a bias magnetic field and operates at the quantum noise level. A specific scheme based on polarization homodyne detection, allows for the generation of ponderomotive squeezing due to the destructive interference between quantum back-action noise (QBA) and photon shot noise (SN). When this oscillator works in the negative mass frame, it can be combined with a source of

entangled photons to suppress quantum noise in a gravitational wave interferometer. In order to do that, the quantum noise frequency response of the spin oscillator should be anti-correlated with the quantum noise contribution in the interferometer output. In this manuscript, the authors also show the potential of achieving this matching by demonstrating a frequency downshift of the spin oscillator virtual rigidity.

The work is scientifically sound and valid, and the authors also present an impressive number of well-accounted calibration techniques (in Methods) to characterize the different quantum noise contributions of the experiment. However, my general impression is that, on its own, this work represents only one important experimental step toward a more complex implementation of the theoretical proposal described in prior works by some of the same authors:

PHYSICAL REVIEW LETTERS 121, 031101 (2018)

PHYSICAL REVIEW D 100, 062004 (2019)

This proposal regards a specific application, which requires a source of quantum light such that correlated beams are sent to the spin oscillator and to the gravitational wave observatory, respectively. The current demonstration of the spin oscillator working in the quantum regime is somehow a step expected by these previous works toward the more complex experimental proposal. Hence, I don't believe the work in its current form can immediately attract the interest of a broader community in the field, as expected for publishing in Nature Communication. Furthermore, the potential application is based on many assumptions that are kind of ideal and hard to implement (e.g. vanishing spin noise and lossless systems). Some theoretical predictions, as those shown in Fig. 7, have been already reported in previous work (PRD above-mentioned). Even the last paragraph in the conclusions, which refers to the broader perspective, is just envisioning the described setup to be coupled to a source of quantum light and a second system such as a second atomic vapor or an optomechanical system, something very hard to implement experimentally and not of direct interest, in my opinion. I would then recommend resubmitting to a more specific journal.

I would have the following questions and suggestions:

1. Is Eq. (1) phenomenological or is it derived somewhere else? The authors should indicate that or a reference, if the expression has been derived from prior art.

2. Is this the first demonstration of ponderomotive squeezing using a simple vapor cell (without cavities) and polarization homodyne detection? The experimental setup looks very simple, with respect to the complex analysis of quantum noise contributions, interference, and suppression. Could the authors comment more on that? Maybe this would deserve another publication alone. A reference about the polarization homodyne detection is needed too.
3. In Fig. 2 e,f it is not clear what is measured, with respect to Figs. 2 c, d. Which is the homodyne angle ϕ in Figs. c and d? Which are the unit of PSD in that same figure? For optimal homodyne phase in Figs. e,f, there is no description of what red and green curves represent, nor to what the reference noise (0 in dB) corresponds to. Should x and y axes units (PSD/linear frequency) be the same in the middle and bottom panels?
4. As described in Fig. 4, why the thermal occupation number and thermal noise increase at lower detection frequencies if the ensemble temperature is the same? The authors should specify this.
5. The normalization of the data in Fig. 3 is quite cryptic, as many points in the manuscript, to the reader not familiar with the subject. The authors say “by normalizing the squeezing spectrum by the transfer function that maps a force acting on the oscillator into the output light” but they don’t provide further explanation. The authors should give more details and maybe an expression of such transfer function, at least in the methods.
6. The authors should indicate the B-field values at different oscillation frequency regimes.
7. In the caption of Fig. 7 the authors refer to a magenta curve, while they refer to a dark red in the text. Is this case even physical (negligible intrinsic linewidth).

Nature Communications manuscript NCOMMS-23-11561

Response to Reviewers' comments

Reviewer #1 (Remarks to the Author):

Reviewer: Atomic spin oscillators are unique systems that allow to mimic mechanical oscillators, at the same time reaching unusual dynamics, such as that of a negative mass oscillator. Atomic spin systems exhibit many of the conventional optomechanical effects, such as ponderomotive squeezing. This finds application in quantum optomechanical systems, where coupling a conventional oscillator to the atomic spin system allows to evade quantum back-action and thus significantly increase the sensitivity or bring mechanical oscillators in unusual quantum states. In [19], some of the authors proposed to cancel quantum back-action in gravitational-wave detectors, thus achieving an enhancement of the sensitivity to gravitational waves. This was proposed as an alternative to a conventional frequency-dependent squeezing generation using nonlinear crystals and large-scale optical filter cavities. The difficulty of achieving this with atomic spin ensemble was in matching the free test mass response in GW detectors. That required reaching a very low oscillation frequency with the spin ensemble, from typical 1MHz down to a few Hz, a very difficult task. In [19] it was proposed that measuring an optimal quadrature on the homodyne detector would introduce a so-called "virtual rigidity", which would utilize quantum correlations between quantum radiation-pressure noise and shot noise to bring the effective frequency down into the required regime. This would still require bringing the oscillators to a lower frequency, but to a more feasible kHz domain. That is what the authors demonstrated experimentally in the present manuscript. Operating the atomic spin system at low frequencies in quantum regime was presented for the first time, in an impressive demonstration of the versatility of the setup, together with a detailed investigation of its various properties, specific to this regime.

The results are scientifically sound, presented with sufficient amount of detail and clarity. The methodology is adequate, and enough detail is provided for reproducing the work. I could not see any flaws in the data analysis, interpretation or conclusions.

Response: We are grateful to the Reviewer for the high valuation of our work. Notably, the revised manuscript includes our latest results where the observation of squeezing and quantum backaction is observed all the way down to sub-kHz frequencies. Those results further enhance the potential effect of our work on the field of quantum sensing in general, and gravitational wave detection in particular.

We have extended the discussion of potential applications in the introduction and in the conclusions. Among those we now highlight biomedical applications of low-frequency magnetometry with sensitivity enhanced by squeezing. Our approach allows to generate low-frequency squeezing using room temperature atomic vapor cell instead of complex and power consuming setups which utilize, e.g., optical nonlinearity.

Reviewer: I do, however, find the presentation to be somewhat lacking. I have two major comments, several minor ones:

1. The authors motivate their work as relevant for gravitational-wave detectors. However, this

motivation looks very focused on a specific narrow application, and even in this application it is far from being a competitor to the existing approaches. It is certainly a very interesting concept, but e.g. Ref. [19] does not present a clear reason for the approach to be practical for GWDs. The benefit there remains lower than the current approaches, and it also arguably lacks in versatility (e.g., as far as I'm aware, it's not applicable to the detuned interferometers). In my view, this weakens motivation, especially when talking about the potential impact. I can imagine there is a broader impact on the field in general, but only very little space is dedicated to it at the very end of the text.

Response: We have expanded the discussion of broader applications of our results in the introduction and in the conclusions. We emphasize the application to contemporary gravitational wave detectors which is even more relevant with our new low acoustic frequency results, signifying the importance of this direction. As a side remark, we note that our approach to broadband noise reduction in GWDs has advantages compared to, for example, the filter cavity approach. In our approach only one of the frequency modes suffers losses in the interferometer whereas the other mode propagates through the low-loss spin ensemble. We thank the reviewer for attracting our attention to the GWD with a detuned interferometer. It will be interesting to explore the applicability of our approach to such interferometers in our future work.

In the updated version of the manuscript, we experimentally prepare the spin oscillator with resonance frequency $|\Omega_s|/(2\pi) = 2$ kHz and $|\Omega_s|/(2\pi) = 1$ kHz with significant contributions of quantum back action noise to the motion. Moreover, we experimentally confirm the downshift of the effective resonance frequency by 2 kHz by means of virtual rigidity from the $\Omega_s=18$ kHz. As stated in the revised conclusion, those two experimental achievements make it possible to emulate the dynamics of the (negative-mass) spin oscillator with effective resonance frequency approaching zero, thus obtaining the free-mass spin oscillator response in the output light, compatible with the GWD response. More detailed investigation, relying on realistic parameters of the atomic ensemble and presented in Fig.5, also predicts promising performance of the protocol, as outlined in the revised manuscript.

2. The overall clarity of the paper could be improved. Many things assume the readers to be familiar with other works by the authors.

Response: We thank the reviewer for these remarks. We have edited the manuscript to make the presentation more clear.

In particular, I'd like to point out a few things:

- The term "virtual rigidity" is never clearly explained. While people in the context of theory behind GW detection know what it meant, it is very unclear for the general audience. It's necessary to explain what is virtual about it, and why it is even called rigidity. I would actually suggest to go away from this term altogether. While a nice analogy, it also is prone to misinterpretation: no real change in the system's dynamics is happening, there's "only" the effect appearing during the measurement.

Response: The term virtual rigidity appears to be quite common in the opto-mechanics community (see, for example, 'Quantum Measurement Theory in Gravitational-Wave Detectors', Stefan L. Danilishin and Farid Ya. Khalili, Living Reviews in Relativity volume 15, 5 (2012)), where in section 4.3 and 4.4 they describe the force normalization of the noise spectrum and the VR effect, respectively.

In the revised manuscript, we explain the effect more carefully and emphasize the virtual nature of the shift; as the reviewer rightly points out, no change occurs in the spin system's dynamics. Importantly, we also explain why the virtual character of the shift is irrelevant to the applications we aim at, namely cases where the spin oscillator plays an intermediary role as a probe system, whereas the actual spin dynamics is not of interest. In view of these arguments and improvements to the text, we hope the referee will agree that the virtual rigidity concept is apt for conveying our work.

Reviewer:

- The central result presented by the authors is the quantum behavior of the system at audio band frequencies. However, the authors never explain (in the main text) how they actually tune the Larmor frequency down to 6kHz. It's implied by its proportionality to B, but it's never clarified. What is new about this frequency? Why was it not possible to go there before? What are the limitations (they are mentioned in the text, but not in a well-structured way)?

Response: The Larmor frequency is proportional to the bias magnetic field, as we clarify on the first and the second pages of the revised manuscript.

In the revised version, we have extended the quantum behavior of the system down to Larmor frequencies below 1 kHz. We have restructured the paper to make the main advances which allow for such low-frequency operation more clear. To summarize, those advances include the reduction of the tensor alignment spin noise which appears at acoustic frequencies close to DC. The reduction is achieved with a higher detuning of the probe light. Moreover, we benefit from a number of technical improvements (redesign of the atomic vapor cell with larger optical depth, minimizing the broadband contribution by probing with a top hat beam, low-noise source of drive magnetic field, shielding, and optimization of spin polarization reducing the impact of detrimental thermal noise).

Reviewer:

- How does the virtual rigidity connect to the observed ponderomotive squeezing level? The authors observed 0.8dB of squeezing at 6kHz, what frequency shift would it translate to with virtual rigidity? Why wasn't this observed in the experiment?

Response: In the revised manuscript we report our latest results demonstrating the key effects at frequencies much lower than those in the original manuscript version. In particular, the revised paper contains a consistent discussion of the new results obtained at the Larmor frequency in the kHz range, including the degree of squeezing and the effect of the virtual rigidity.

In general, the shift imposed by virtual rigidity doesn't directly depend on the level of ponderomotive squeezing. The size of the frequency shift rather depends on the ratio between the readout rate Γ_s and the initial resonance frequency Ω_s as well as on the chosen homodyne phase ϕ . However, virtual rigidity is (indirectly) linked to the level of ponderomotive squeezing, since both phenomena are defined by the readout rate Γ_s .

In the revised manuscript we set the resonance frequency to $|\Omega_s|/(2\pi)=18\text{kHz}$ and present both 5.5 dB noise reduction below shot noise and the 2.1 kHz downshift of the effective Larmor frequency by virtual rigidity. In the lower audio band (down to 1 kHz) we observe smaller ponderomotive squeezing due to technical noise. In this domain we can expect a frequency downshift by virtual rigidity which is larger than the spin oscillation frequency conditioned on maintaining the same readout rate. We have chosen not to show the corresponding data in the manuscript due to complications related to the presence of laser intensity noise and other DC noise sources.

Reviewer:

- There's a lot of incoherence between the different parts of the paper. E.g. sometimes back-action

noise is called QBA, sometimes QBAN, sometimes BAN. Spectral densities have different normalizations, which are not very well explained. What's the difference between PSD[dB] and PSD[SN]? Sometimes virtual rigidity is called virtual spring.

Response: We thank the reviewer for noting those inconsistencies which are eliminated in the revised manuscript. The abbreviation QBA refers to the (quantum back action) phenomenon, whereas QBAN (quantum back action noise) refers to the associated noise induced by this mechanism. Both abbreviations are explained and used in the text, depending on the context. The term 'virtual rigidity' refers to the (virtual optical) spring softening, and both names are used throughout the text. As explained in the text, the label PSD [SN] means power spectral density in shot noise units, but in some plots the logarithmic scale is used (denoted as PSD/SN [dB]).

Reviewer:

- The structure of the paper is not very clear. The authors announce the oscillators in acoustic band, but make the first part about the MHz band (which were published elsewhere before). Virtual rigidity appears in the next section, and it's again for MHz oscillators, while it would be most relevant for the kHz ones, if the paper tries to align it with proposal in [19]. Most of the rest of the paper is about observation of quantum features at audioband frequencies and technical issues associated with that, and it's not clear how it connected to observing virtual rigidity. I understand, that the intention is to say that virtual rigidity is demonstrated to be on the kHz level, and thus shifting the Larmor frequency down to kHz level would allow to make the effective frequency close to zero. This is hard to understand from text.

Response: The revised manuscript addresses those concerns of the reviewer in full, as both the virtual rigidity and squeezing are now observed in the audio band.

Reviewer:

This all makes the paper quite hard to follow. I would recommend the authors to give a brief introduction to their main results already in the beginning of the paper, possibly extending the focus from GWDs to a broader picture (in the view of my point 1). Possibly, there exists a better structure for the paper (e.g. start with audioband observations to show that a low-frequency oscillator is possible, and then talk about what kind of virtual rigidity is required, and then demonstrate its existence for a high-frequency system).

Response: We have followed the recommendations of the reviewer and have restructured the manuscript to make it clearer.

Reviewer: Overall, I find the results to be quite beautiful. They deserve to be seen by a broad community and to make an impact on experiments and theoretical investigation. They would be appropriate for publication in a high-ranking journal, such as Nature Communications, but the current presentation needs to be improved.

Response: We are grateful to the Reviewer for recommending publication of our work and have made every effort to make the presentation clearer following the Reviewer's advice.

Reviewer: Here're some minor comments:

1. On Fig.2 the squeezing is 4.9dB, in the text it's 4.8db.

Response: the inconsistency has been eliminated (the discussion of spin oscillator performance at 1 MHz was moved to methods together with Fig.2)

2. In the same place it's stated that the squeezing for the negative mass is only slightly reduced. In practice it's more than 1dB difference, which is quite significant. Also, isn't it the squeezing for the negative mass that is relevant? Then it's somewhat misleading.

Response: New results are included in the revised manuscript demonstrating squeezing and quantum backaction all the way down to the sub-kHz frequency range, while presenting the negative-mass configuration in greater detail.

Reviewer:

3. Fig.2 (e,f) is not color coded. Some legend would be useful.

Response: Fig. 2 has been completely revised and now presents the new squeezing and virtual rigidity data obtained in the acoustic frequency range. Color coding and a detailed legend facilitate the explanation of the results.

4. Fig.3 lacks units on the y axis. Also it should be explained more clearly why another normalization is used (it's mentioned in the text, but not explained).

Response: the exact procedure of normalization required for demonstration of virtual rigidity is now quantitatively explained in the main text and outlined in more detail in the Supplementary section. Units are shown.

5. In Fig.5, the meaning of the red shaded area in (c) is not clear.

Response: Fig.5 [c], describing the influence of the laser intensity noise on the level of ponderomotive squeezing, is removed.

6. In Fig. 7, the meaning of the yellow circles & arrow is not clear (is it a representation of the spin system? May be then simply labeling it as Ω_s is sufficient).

Response: the yellow (orange) circles & arrow indicate the initial Larmor frequency of the spin ensemble, which is optimal for the protocol. The higher starting Larmor frequency can be set (since 2 kHz downshift by virtual rigidity is experimentally demonstrated in the present experiment), but such choice will impair the quantum noise reduction (as explained in PHYSICAL REVIEW D 100, 062004 (2019)).

We thank the reviewer for careful reading of the manuscript and noting inconsistencies, which we have addressed in full in the revised manuscript.

Reviewer #2 (Remarks to the Author):

Reviewer: This work reports the study of noise spectrum of atomic spin oscillator in the acoustic band, and the pondermotive squeezing and back-action reduction is demonstrated down to 6kHz. This is a notable technical achievement, as any experimentalist knows the Sisyphus nature of $1/f$ noise. The motivation and focus of current study seem to be mostly on potential applications for gravitational wave detection, although I think the paper would benefit from a more extensive discussion of other potential applications that may benefit from low-frequency quantum noise control. I find the presented results interesting and potentially impactful for quantum sensor community.

Response: We are grateful to the Reviewer for finding our results interesting and impactful. Notably, the revised manuscript includes our latest results where observation of squeezing and subsequent quantum backaction is observed all the way down to sub-kHz frequencies. Those results further enhance the potential effect of our work on the field of quantum sensing. We have extended the discussion of potential applications in the introduction and in the conclusions. We still emphasize the quantum-enhanced metrology application for contemporary gravitational wave detectors (GWD), signifying the importance of this direction. Apart from that, we highlight biomedical applications of low frequency magnetometry and bio samples trapping & tacking with sensitivity enhanced by squeezing. We stress that our approach allows to generate low-frequency squeezing using room-temperature atomic vapor instead of complex and power consuming setups which utilize, e.g., optical nonlinearity.

Reviewer: However, I found the paper occasionally hard to follow due to some peculiarities of writing style and language. For example, I am not sure what is meant by the term "imprecision noise" used several time in the manuscript. Fig.2 uses notation "BAN" and "TH" which I presume mean back-action noise and thermal noise, but in the text QBA abbreviation is used, and TH is not explained. Also, I must confess I had particularly hard time following the section "Frequency downshift of the spin oscillator by virtual rigidity". I am not an expert in ophomechanics, but the heavy use of specialized therminology made it really hard for me to understand the relevance of this discussion to the rest of the manuscript. I think a more streamlined presentation with clearly formulated objectives would benefit readers who are, like me, may benefit from this work but not necessarily fully immersed in the details of the atomic spin oscillator performance.

Response: The inconsistencies in notations are eliminated, more comments on virtual rigidity are added and the presentation is streamlined following the Reviewer's suggestions. More justifications for the importance of our results have been added in the introduction and conclusions.

Reviewer #3 (Remarks to the Author):

In this manuscript, Jia et al. report an experimental demonstration of an atomic ensemble in a vapor cell, which behaves as a spin oscillator in the presence of a bias magnetic field and operates at the quantum noise level. A specific scheme based on polarization homodyne detection allows for the generation of pondermotive squeezing due to the destructive interference between quantum back-action noise (QBA) and photon shot noise (SN). When this oscillator works in the negative mass frame, it can be combined with a source of entangled photons to suppress quantum noise in a gravitational wave interferometer. To do that, the quantum noise frequency response of the spin

oscillator should be anti-correlated with the quantum noise contribution in the interferometer output. In this manuscript, the authors also show the potential of achieving this matching by demonstrating a frequency downshift of the spin oscillator virtual rigidity. The work is scientifically sound and valid, and the authors also present an impressive number of well-accounted calibration techniques (in Methods) to characterize the different quantum noise contributions of the experiment.

Response: We are grateful to the Reviewer for the high valuation of our work.

Reviewer: However, my general impression is that, on its own, this work represents only one important experimental step toward a more complex implementation of the theoretical proposal described in prior works by some of the same authors:

PHYSICAL REVIEW LETTERS 121, 031101 (2018)

PHYSICAL REVIEW D 100, 062004 (2019)

This proposal regards a specific application, which requires a source of quantum light such that correlated beams are sent to the spin oscillator and to the gravitational wave observatory, respectively. The current demonstration of the spin oscillator working in the quantum regime is somehow a step expected by these previous works toward the more complex experimental proposal. Hence, I don't believe the work in its current form can immediately attract the interest of a broader community in the field, as expected for publishing in Nature Communication. Furthermore, the potential application is based on many assumptions that are kind of ideal and hard to implement (e.g. vanishing spin noise and lossless systems).

Response: In the revised manuscript we made an effort to highlight applications of our results beyond gravitational wave interferometry.

Importantly, the revised paper contains our most recent experimental results where we have extended the observed squeezing and quantum back action for the negative-mass spin oscillator down to the sub-kHz region. In the revised version, we confirm the strong contribution of QBAN to the spin dynamics at the lower end of the acoustic range (in particular, for $|\Omega_s|/(2\pi)=2$ kHz and $|\Omega_s|/(2\pi)=1$ kHz). Moreover, we demonstrate the downshift of an effective Larmor frequency by 2 kHz. We believe that these two experimental results obtained for the negative-mass configuration prove the possibility to reach the required regime of virtual free-mass response with our system. We have thus demonstrated that our system is highly promising for noise reduction in the acoustic band, including gravitational wave detection.

We have extended the discussion of potential applications in the introduction and in the conclusions. Among those we now highlight biomedical applications for low-frequency optically-pumped magnetometry with sensitivity enhanced by pondermotive squeezing.

Besides the GWD and optically-pumped magnetometry application, the reported results make our setup a unique source of quantum light in the audio frequency domain. Compared to standard approaches for generation of squeezing using parametric down conversion, our approach is profoundly simpler and much less power consuming, which makes it more practical and versatile.

Reviewer: Some theoretical predictions, as those shown in Fig. 7, have been already reported in previous work (PRD above-mentioned). Even the last paragraph in the conclusions, which refers to the broader perspective, is just envisioning the described setup to be coupled to a source of quantum light and a second system such as a second atomic vapor or an optomechanical system,

something very hard to implement experimentally and not of direct interest, in my opinion. I would then recommend resubmitting to a more specific journal.

Response:

We would like to emphasize that Fig.5 (revised Fig.7 from the initial manuscript) is plotted for a realistic configuration of parameters. In particular, the entanglement source required for parallel probing of a GWD and of the reference negative-mass spin system has been reported in our paper '*Two-colour high-purity Einstein-Podolsky-Rosen photonic state*', *Nature Comm.* 2022.

Coming back to parameters of the atomic ensemble, we show that deviation from the configuration with vanishing extraneous spin thermal noise (i.e., the spin noise above the projection noise level)_z will not preclude us from demonstrating broadband sensitivity improvement of contemporary GWDs (the case of thermal occupation $n_s=2$ is shown in the figure). The detrimental effect of optical losses is also studied in PHYSICAL REVIEW D 100, 062004 (2019). Based on that work, we estimate that $\leq 10\%$ losses for each channel achieved in the Nature Comm (2022) and the reported experiment, are still acceptable for significant quantum noise reduction in GWDs.

Reviewer:

I would have the following questions and suggestions:

1. Is Eq. (1) phenomenological or is it derived somewhere else? The authors should indicate that or a reference, if the expression has been derived from prior art.

Response: we added the proper references in the revised manuscript.

Reviewer:

2. Is this the first demonstration of ponderomotive squeezing using a simple vapor cell (without cavities) and polarization homodyne detection? The experimental setup looks very simple, with respect to the complex analysis of quantum noise contributions, interference, and suppression. Could the authors comment more on that? Maybe this would deserve another publication alone. A reference about the polarization homodyne detection is needed too.

Response:

We thank the Reviewer for the high valuation of our ponderomotive squeezing results. To the best of our knowledge such results have not been published to date. In fact, observation of ponderomotive squeezing in the acoustic frequency band is one of the main points of the paper. We prefer to report those results in this manuscript, rather than dedicate another publication to it. Following the Reviewer's comments, we have made this point clearer in the revised manuscript. Despite the seemingly simple experimental setup, demonstration of such squeezing in the acoustic band presented in our paper required several critical advances, such as redesigning the cell geometry providing larger optical depth, using the top-hat probe beam, minimization of the impact of the tensor alignment spin noise ('DC noise'), careful optimization of the spin polarization for reduction of thermal spin noise, utilization of ultralow noise and homogeneous sources of external magnetic field with PCB coils system and proper shielding.

The relevant references for polarization homodyne detection have been added.

Reviewer:

3. In Fig. 2 e,f it is not clear what is measured, with respect to Figs. 2 c, d. Which is the homodyne

angle ϕ in Figs. c and d? Which are the unit of PSD in that same figure? For optimal homodyne phase in Figs. e,f, there is no description of what red and green curves represent, nor to what the reference noise (0 in dB) corresponds to. Should x and y axes units (PSD/linear frequency) be the same in the middle and bottom panels?

Response: The structure of the paper is profoundly revised to make the main messages clearer. The new results in the audio frequency range are central in the revised manuscript, while the discussion of the spin oscillator in the MHz range (including Fig.2) for the effective mass calibration is now moved to the Methods (Sec.E). We have fixed all inconsistencies and added missing information.

Reviewer:

4. As described in Fig. 4, why does the thermal occupation number and thermal noise increase at lower detection frequencies if the ensemble temperature is the same? The authors should specify this.

Response: The ‘true’ thermal occupation should be the same for any resonance frequency of the spin oscillator since we don’t expect any dependence of the spin polarization on the Larmor frequency. Nevertheless, while fitting the spectra with the spin noise model Eq. (1), we extract an increased n_s at lower Ω_s . As a possible explanation, we point at the incompleteness of the model Eq. (1) close to the DC frequency range. In particular, we identify the spin response, centered at zero frequency and presumably originating from tensor alignment spin dynamics (although, other noise sources like laser classical intensity noise are also possibly present). This is outlined in the manuscript, and a detailed investigation of the tensor spin noise at DC frequencies is provided. Such detrimental noise is now incorporated into n_s which now plays the role of an *effective* thermal occupation. Fig.4 is revised since the new results are included, but the tendency of increased thermal occupation for lower frequencies persists.

Reviewer:

5. The normalization of the data in Fig. 3 is quite cryptic, as many points in the manuscript, to the reader not familiar with the subject. The authors say “by normalizing the squeezing spectrum by the transfer function that maps a force acting on the oscillator into the output light” but they don’t provide further explanation. The authors should give more details and maybe an expression of such transfer function, at least in the methods.

Response: The exact procedure of normalization with subsequent demonstration of the effective frequency downshift by virtual rigidity is now outlined in the Supplementary section using input-output relations for quadratures instead of power spectral density. To summarize, we need to extract the light noise (QBAN + SN + cross-correlations) normalized by the transfer function of the thermal force (the factor N_{th} in Supplementary) into the output light.

Reviewer:

6. The authors should indicate the B-field values at different oscillation frequency regimes.

Response: In the revised version we give the scaling ratio between applied magnetic field and resulting Larmor frequency (page 2) and clearly state the value Ω_s for each outlined case.

7. In the caption of Fig. 7 the authors refer to a magenta curve, while they refer to a dark red in the text. Is this case even physical (negligible intrinsic linewidth).

Response: In the current Fig.5 (previously Fig.7) the magenta curve corresponds to the quantum noise limited Interferometer, whereas the dark curve predicts the quantum noise suppression when GWD is coupled with a negative mass spin ensemble by means of entangled light. The sub-Hz intrinsic linewidth, being feasible for the vapor cells with larger cross section areas and high quality of anti-relaxation coating [1], would be sufficiently smaller than the power broadening part of the spin decay rate for the required atomic readout rate (the latter is predominantly defined by the probe optical power).

We thank the Reviewer for constructive criticism. We hope that the Reviewer agrees that the revised paper containing new results meets the standards of Nature Communications.

[1]

Mikhail Balabas [Polarized alkali-metal vapor with minute-long transverse spin-relaxation time]
Phys. Rev. Lett. 105, 070801 (2010)

REVIEWER COMMENTS

Reviewer #1 (Remarks to the Author):

The updated manuscript mostly addresses the concerns I raised in my first review. The clarity of the manuscript is significantly improved. I still have a few comments left and recommend a revision.

1) In fig.2(a), the blue shaded area represents thermal noise, and it's stated that the fit of this curve gives the parameters. However, if I understand it correctly from the text, the fits of the curves in (b) give the actual information on the parameters. If so, this should be reflected in the caption. Otherwise, the text should be clarified.

2) I didn't understand the explanation for the difference in n_{eff} between the fit and MORS theory. It's stated that laser noise might be responsible for the added occupation. Does that imply that the noise in Fig.1(a) is actually not the thermal noise, but a mixture of thermal and classical radiation-pressure noise? I suggest the authors to clarify this in the text.

3) I find the structure of the part on frequency downshift still somewhat confusing. The reader who is not familiar with the term will not understand the first paragraph. The term "virtual rigidity" is not common (in fact, it's only used by a couple of people coming from Braginsky group, as far as I know). It does reflect nicely the observed behavior here, but it needs better connection to more well-known terminology. For example, the authors could state the problem first: that achieving low-frequency sensors with specific response is very important for application X,Y,Z, but achieving such naturally is very difficult (and state why). At the same time, there's a way to post-process the data without modifying the response of the system itself, by using correlations between the light fields, such that the newly defined system has a required response. Basically, it's simply re-defining the variables of the system with correlations such that it looks like the system without correlations but with a shifted frequency. Writing it in a (similarly) simple manner would greatly help understanding of the concept. Of course, it can be followed by the more technical explanation as it is given now.

4) Is there an intuitive physical picture for the DC-noise? It might be helpful to present one.

Of course, I realize there might not be one, but encourage the authors to consider it.

5) In Fig.4, the size of the ticks and labels is much larger than for the other plots.

6) In discussion, it should be clarified that combination of the 2kHz oscillator with virtual optical spring still has to be demonstrated.

7) Also, the influence of the DC noise impacted the sensitivity already at kHz. Is it feasible to assume that it could be reduced at lower frequencies? The authors should comment on that.

8) For Fig.5, the authors use the parameters that are more optimistic compared to the presented ones: smaller thermal noise, higher cooperativity, no DC noise, much lower frequency of the spin oscillator. With these parameters the sensitivity improvement would probably be comparable (or even less) than a conventional approach. I realize that the authors have to make a projection based on their current results and knowledge, but it would be good to clarify these points, otherwise it sounds like the approach is quite ready to be implemented in GW detectors, which I believe is not yet the case.

9) Importantly, nowhere in the text it's stated that the proposal for GW detectors is an alternative to the existing approach of frequency-dependent squeezing, which is already implemented in the current detectors (and helps detecting GWs right now). This should be clarified, otherwise the readers might be misled to think that this is the only approach that would bring huge benefit to GW detection. In fact, there are many different ones, each promising comparable improvement in sensitivity, only differing by the complexity and cost of the setup. The authors should a statement on that.

10) In fact, I see the strength in application to other sensors, as authors point out further in the text. For sub-kHz squeezed light, it would be good to give an estimate on what levels of squeezing the authors envision to be possible in the improved setup.

Reviewer #3 (Remarks to the Author):

I would like to thank the authors for following my suggestions and satisfactorily responding to my comments. I think that the revised manuscript has been significantly improved and it could now meet the standards of Nature Communications. However, before recommending publication, I would like to see the last following suggestions implemented in the final version:

1. The authors have now included biomedical applications like those based on optically pumped magnetometers among the extended potential applications of the ponderomotive squeezing they demonstrated. However, it seems that optimal squeezing reported by the authors is obtained for a very specific choice of polarization angle and phase in the homodyne detection, where the latter is set by two wave plate angles. In optical magnetometers, a balanced detector at fixed half-waveplate angle is usually employed for polarization rotation measurements, or near-resonant absorption measurements of a circularly polarized pump/probe beam are performed. Could the authors comment on the compatibility of their squeezing technique with the detection schemes used in optical magnetometry?

2. The authors should cite C. Troullinou, R. Jiménez-Martínez, J. Kong, V. G. Lucivero, and M. W. Mitchell, Phys. Rev. Lett. 127, 193601 (2022), either in the introduction about back-action evading measurements or in the conclusions about applications to magnetometry.

3. It is not specified that Eq. (1) is the “noise” power spectral density, and not the simple PSD. In fact, there is no signal term and it is all about noise contributions. Could the author better specify this?

4. PSD[SN] in the y-axis of Figs. 2 and 4 is still confusing, I think other reviewers had the same remark. I would suggest using the real PSD units, V^2/Hz or maybe rad^2/Hz ?

5. There is no comment about spin squeezing, while the implemented setup and QND light-atoms interaction Hamiltonian looks like an ideal configuration to generate spin squeezing too. Could the authors comment on this aspect? Is the generated squeezing only sub-shot-

noise light squeezing?

6. The authors should cite V. G. Lucivero, N. D. McDonough, N. Dural, and M. V. Romalis, Phys. Rev. A 96, 062702 (2017) when describing the effects of atomic diffusion on spin noise spectra in Section B of Methods.

Reviewer #1 (Remarks to the Author):

The updated manuscript mostly addresses the concerns I raised in my first review. The clarity of the manuscript is significantly improved.

Response:

We are grateful to the reviewer for recognizing the enhanced quality of our revised manuscript.

I still have a few comments left and recommend a revision.

1) In fig.2(a), the blue shaded area represents thermal noise, and it's stated that the fit of this curve gives the parameters. However, if I understand it correctly from the text, the fits of the curves in (b) give the actual information on the parameters. If so, this should be reflected in the caption. Otherwise, the text should be clarified.

Response:

We thank the Reviewer for noticing the inconsistency between the description of the fit procedure between the caption of Figure 2 and the text. In fact, we perform the fit of several traces recorded at different homodyne detection angles. For example, we select the phase quadrature and the quadrature giving the strongest ponderomotive squeezing, corresponding to Fig.2a and Fig2b, respectively, as stated in the current version of the text. We then optimize the fits and extract the common parameters such as readout rate, linewidth and thermal occupation.

The caption for Fig.2 has been changed such that it refers to the text in order to explain the procedure of the fit (meaning reconstruction of the QBAN and TN, which are displayed on Fig.2[a]). The sentence in the caption now reads as:

“The fitting of experimental traces using noise model Eq.(1) is described in the text. Reconstructed quantum back-action noise (QBAN) and thermal noise (TN, defined by thermal occupation $n_S = 3.5$) are shown as the light red shaded area and the light blue shaded area, respectively. The ratio between QBAN and TN results in the quantum cooperativity $C_q = 3$.”

2) I didn't understand the explanation for the difference in n_{eff} between the fit and MORS theory. It's stated that laser noise might be responsible for the added occupation. Does that imply that the noise in Fig.2(a) is not the thermal noise, but a mixture of thermal and classical radiation-pressure noise? I suggest the authors clarify this in the text.

Response:

We thank the Reviewer for raising this question. The answer might be divided into two parts.

First, we indeed regard the thermal occupation obtained from the fitting with the full spin noise model (eq.1) in Fig2 (a) as an 'effective' n_S , which includes noise sources that are not captured by Eq.1. Such an

assumption remains reasonable in the context of the atomic response within a narrow band range. We utilize the cross-validated 'actual' thermal occupancy sourced from MORS and the spin noise spectrum recorded at 1 MHz to quantitatively evaluate the additional noise at low Larmor frequency. In order to clarify this circumstance, we have added the paragraph to the end of section C of the Methods which now reads:

“Using Eq.(1) to fit the spectrum taken at $\Omega S / 2\pi = 18$ kHz, we obtain the thermal occupation $nS = 3.5$ which is larger than the result obtained from the calibration by MORS in the continuous regime. The nS extracted from the full spin model is then treated as an effective thermal occupancy that includes additional noise sources not accounted for in Eq.(1), for example, intensity noise of the probe laser.”

Secondly, we do consider the possibility that the intensity noise of the probe laser is perceived by the atomic spin ensemble within a framework of Faraday interaction model. Such a contribution to the total noise budget can be treated as ‘classical’-back action noise. It might also be taken into account as an increase in the (effective) thermal occupation. However, this hypothesis has to be thoroughly investigated. Until this is done, we prefer to avoid using the expression ‘classical back-action noise’ in the manuscript. From an alternative point of view, we envision the impact of noise sources other than the intensity noise of the probe laser (for example, atomic noise at DC and stray magnetic noise coming from the environment).

3) I find the structure of the part on frequency downshift still somewhat confusing. The reader who is not familiar with the term will not understand the first paragraph. The term "virtual rigidity" is not common (in fact, it's only used by a couple of people coming from Braginsky group, as far as I know). It does reflect nicely the observed behavior here, but it needs better connection to more well-known terminology. For example, the authors could state the problem first: that achieving low-frequency sensors with specific response is very important for application X,Y,Z, but achieving such naturally is very difficult (and state why). At the same time, there's a way to post-process the data without modifying the response of the system itself, by using correlations between the light fields, such that the newly defined system has a required response. Basically, it's simply re-defining the variables of the system with correlations such that it looks like the system without correlations but with a shifted frequency. Writing it in a (similarly) simple manner would greatly help understanding of the concept. Of course, it can be followed by a more technical explanation as it is given now.

Response:

Following the Reviewer's advice, we have removed the term “virtual rigidity” and substituted it with a more explicit expression “virtual frequency downshift”. We explicitly state the incentive for studying this effect on the first page of the paper as follows “... the virtual oscillator frequency downshift, which is, for example, necessary for matching the spin response to that of a GWD [23] as well as for other sensing applications in the acoustic frequency range [26]” where we have added a reference to low frequency magnetic sensing. Following the Referee's suggestion, we now introduce the virtual frequency shift in the paragraph below Eq. (1) of the manuscript (the text starting with “The term containing...” This is followed by a detailed explanation in the first paragraph in “Virtual frequency downshift of the observed oscillator response” which contains an intuitive picture of this effect and its role in the present work and its applications.

4) Is there an intuitive physical picture for the DC-noise? It might be helpful to present one. Of course, I realize there might not be one, but encourage the authors to consider it.

Response:

As stated on p.4 in the main text and in the section F2 in the Methods, we understand the ‘DC noise’ as the noise arising from the dynamics of spin alignment operators that (in particular) leads to a component centered at zero frequency in the spin noise spectra. This DC noise could potentially arise due to imperfect optical pumping, particularly in situations where strong probe light is present. Additionally, it is possible that the classical intensity fluctuations of the probe field may further amplify this DC noise.

5) In Fig.4, the size of the ticks and labels is much larger than for the other plots.

Response:

We thank the reviewer for pointing out the difference, figure 4 has now been resized.

6) In discussion, it should be clarified that combination of the 2kHz oscillator with virtual optical spring still has to be demonstrated.

Response:

Following the Reviewer’s comment, the last sentence on p. 5 is modified and reads “...we expect to emulate the motion of a free-mass object...” to avoid misunderstanding. The technical difficulty of demonstrating the downshift of the effective resonance frequency in the lower audio band is elaborated in the Supplementary (the last paragraph) of the revised version of the manuscript, which now reads as

“To observe the effective shift of the atomic resonance frequency, one can extract the (symmetrized) spectrum $\langle f_{\text{LN}}^{\dagger} f_{\text{LN}} + f_{\text{LN}} f_{\text{LN}}^{\dagger} \rangle / 2$ of the light force from experimental data. We implement the procedure for the spin oscillator in the upper audioband ($\Omega S / 2\pi = 18$ kHz) and demonstrate the frequency downshift $|\Delta\Omega S / 2\pi| \approx 2.1$ kHz, as shown in Fig. 2[c]. In the frequency range $\lesssim 3$ kHz, the model Eq. (1) no longer accurately describes the noise budget, due to the more pronounced impact of DC noise. This prevents the demonstration of a quantum-limited spin oscillator with effective frequency in this range in the present experimental setup.”

7) Also, the influence of the DC noise impacted the sensitivity already at kHz. Is it feasible to assume that it could be reduced at lower frequencies? The authors should comment on that.

Response:

We expect that the reduction of intensity noise of the probe laser in the lower acoustic range will facilitate the mitigation of ‘DC-noise’. At the same time, there could be extra noise sources, unrelated to the dynamics of tensor alignment spin operators but being disguised as ‘DC-noise’. The role of power

fluctuations of the probe laser compared to other alleged noise sources remains unclear until an intensity stabilization loop is implemented.

8) For Fig.5, the authors use the parameters that are more optimistic compared to the presented ones: smaller thermal noise, higher cooperativity, no DC noise, much lower frequency of the spin oscillator. With these parameters the sensitivity improvement would probably be comparable (or even less) than a conventional approach. I realize that the authors have to make a projection based on their current results and knowledge, but it would be good to clarify these points, otherwise it sounds like the approach is quite ready to be implemented in GW detectors, which I believe is not yet the case.

Response:

We have modified the text (section 'Discussions', 2nd paragraph, last 2 sentences, p.6) and specified the main limitations of the atomic spin ensemble reported in the manuscript. The parameters used to produce Fig.5 are explicitly outlined in the text (p.6) and can be compared to the features of the spin oscillator demonstrated in the experiment (such as readout rate, cooperativity, thermal occupation and (the lowest) resonance frequency of the atomic oscillator).

9) Importantly, nowhere in the text is it stated that the proposal for GW detectors is an alternative to the existing approach of frequency-dependent squeezing, which is already implemented in the current detectors (and helps detecting GWs right now). This should be clarified, otherwise the readers might be misled to think that this is the only approach that would bring huge benefit to GW detection. In fact, there are many different ones, each promising comparable improvement in sensitivity, only differing by the complexity and cost of the setup. The authors should make a statement on that.

Response:

We have added relevant references (section 'Discussion', 2nd paragraph, the last sentence, p.6) and acknowledged the efforts towards the broadband quantum noise suppression by injecting frequency-dependent squeezed vacuum, while highlighting possible advantages of the negative-mass spin oscillator over filtering resonators.

The newly included discussion is now presented as

“The reduction of the intrinsic atomic linewidth $\gamma_{S,0}$ together with the mitigation of ‘DC-noise’ will make it possible to reach a sensitivity improvement of GWDs comparable to the predicted performance of other quantum-noise-evasion protocols [41]. The advantages of our approach in comparison to, e.g., achieving frequency-dependent squeezing by means of a long filter cavity [42, 43], include the tunability of the quantum noise evasion (via Γ_S , Ω_S and Φ) and its small physical footprint. Another possible advantage is the reduced effect of optical losses in the GWD, which is due to the fact that only one of the two entangled modes propagates in the GWD, whereas the other mode interacts with the relatively low loss spin ensemble [22, 23].”

10) In fact, I see the strength in application to other sensors, as authors point out further in the text. For sub-kHz squeezed light, it would be good to give an estimate on what levels of squeezing the authors envision to be possible in the improved setup.

Response:

This question appears to be closely related to the issues raised in Reviewer's comment NO.7. At the present time, we have not yet developed the ability to effectively model or mitigate the 'DC-noise', Once we overcome this obstacle (we are working on an active feedback system to reduce the classical intensity noise), the prediction of expected squeezing level in the upgraded setup will be feasible. As of now, the complex (and unknown) structure of low frequency noise makes it hard to give a more precise response at this time.

Reviewer #3 (Remarks to the Author):

I would like to thank the authors for following my suggestions and satisfactorily responding to my comments. I think that the revised manuscript has been significantly improved and it could now meet the standards of Nature Communications.

Response:

We thank the reviewer for recognizing the considerable enhancements in the quality of our revised manuscript.

However, before recommending publication, I would like to see the last following suggestions implemented in the final version:

1.The authors have now included biomedical applications like those based on optically pumped magnetometers among the extended potential applications of the ponderomotive squeezing they demonstrated. However, it seems that optimal squeezing reported by the authors is obtained for a very specific choice of polarization angle and phase in the homodyne detection, where the latter is set by two wave plate angles. In optical magnetometers, a balanced detector at fixed half-waveplate angle is usually employed for polarization rotation measurements, or near-resonant absorption measurements of a circularly polarized pump/probe beam are performed. Could the authors comment on the compatibility of their squeezing technique with the detection schemes used in optical magnetometry?

Response:

We thank the Reviewer for prompting us to better explain the application of our source for sensing. For the magnetometry applications that we discuss in the conclusions we propose applying our setup as a source of input squeezed light for a magnetometer. The light coming out of our source after the quarter wave and half wave plates is squeezed in the difference between two linear polarization components which is detected in our experiment by polarization beam splitter and balanced detection. As the Referee points out, polarization rotation is indeed a typical signal in optical magnetometry. Therefore, if a sample

whose polarization rotation is to be measured is placed between the halfwave plate and the polarizer in Fig. 1, this polarization rotation will be detected with sub-shot noise sensitivity provided by squeezed light. As the central frequency of the reported source of squeezing is tunable by magnetic field, it can be adjusted to the ac polarization rotation signal to be detected. Note that applications of such relatively simple squeezed source are not limited to magnetometry, as it can be also used for other kinds of sub shot noise interferometry. We have added a sentence to the conclusion elucidating the above points.

2. The authors should cite C. Troullinou, R. Jiménez-Martínez, J. Kong, V. G. Lucivero, and M. W. Mitchell, Phys. Rev. Lett. 127, 193601 (2022), either in the introduction about back-action evading measurements or in the conclusions about applications to magnetometry.

Response:

We thank the reviewer for providing the reference, which ref. [47] has now been added to the conclusion of the revised manuscript. The sentence now reads as follows

“The robust and tunable squeezed light source reported here is relevant for quantum magnetometry [47], especially for biomedical applications where signals in the sub-kHz range often prevail”

3. It is not specified that Eq. (1) is the “noise” power spectral density, and not the simple PSD. In fact, there is no signal term, and it is all about noise contributions. Could the author better specify this?

Response:

To address this concern, we would like to emphasize that the primary goal of our study is to carefully calibrate the noise of the macroscopic atomic spin oscillator and explore the applicability of broadband quantum noise reduction for GWD systems based on our current results. Therefore, we exploit Eq.(1) that describes the total noise budget but doesn't contain any 'signal' term in the model of atomic spin (as would be the case for i.e., magnetometry). We believe that it is clearly specified in the current version of the manuscript.

The observation of dominated quantum backaction noise in the negative mass oscillator, accompanied by the compelling evidence of pondermotive squeezing of light, would constitute an important step towards the broadband noise reduction in gravitational wave detectors. In the context of this proposal [4], gravitational signals are acquired through gravitational wave detectors such as LIGO and Virgo. To investigate the quantum noise reduction in GWD coupled to the atomic ensemble, one will have to add the respective signal term and introduce the expression for the(strain) sensitivity. However, this would bring us beyond the scope of the present manuscript.

4. PSD[SN] in the y-axis of Figs. 2 and 4 is still confusing, I think other reviewers had the same remark. I would suggest using the real SD units, V^2/Hz or maybe rad^2/Hz ?

Response:

The power spectral density (PSD) of light probing the atomic ensemble was initially acquired in V^2/Hz units. While doing the data analysis, we normalize the measured light noise with respect to the averaged optical shot noise (SN) and subsequently fit it to the spin noise model outlined in Eq. (1). As a result, the traces on Fig.2 and Fig.4 represent dimensionless PSD in SN units.

We would like to maintain the shot noise normalization (as was done on Fig.2 and Fig.4 in the current version of the manuscript) to ensure consistency with the units used in the theoretical model. Additionally, presenting the shot noise unit in a decibel scale allows for better visualization of the ponderomotive squeezing phenomenon. However, to improve reader comprehension and ensure clarity, we have left the comments in the caption of Figures.

5. There is no comment about spin squeezing, while the implemented setup and QND light-atoms interaction Hamiltonian looks like an ideal configuration to generate spin squeezing too. Could the authors comment on this aspect? Is the generated squeezing only sub-shot-noise light squeezing?

Response:

We appreciate the reviewer's insightful observation regarding the generation of spin squeezing. The squeezing demonstrated in our work is indeed squeezing of light quadrature. Meanwhile, it is possible to produce conditional spin squeezing when the information of QND Faraday rotation measurement of atomic spin (especially in the strong coupling regime, meaning $Cq \gg 1$) is used to predict and reduce its conditional variance. Nonetheless, our goal is to present a quantum backaction noise dominant macroscopic spin oscillator operating in acoustic regime, which can potentially help to reduce the quantum noise in gravitational wave detection system. Therefore, we decided not to focus on the conditional spin squeezing in our current work.

6. The authors should cite V. G. Lucivero, N. D. McDonough, N. Dural, and M. V. Romalis, Phys. Rev. A 96, 062702 (2017) when describing the effects of atomic diffusion on spin noise spectra in Section B of Methods.

Response:

We thank the reviewer for providing the reference, which has now been added to the broadband noise discussion in section E.

REVIEWERS' COMMENTS

Reviewer #1 (Remarks to the Author):

All my comments were addressed to my satisfaction and I'm happy to recommend the manuscript for publication.

Reviewer #3 (Remarks to the Author):

I thank the authors for satisfactorily following my last suggestions and recommendations. I think that the revised manuscript now meets the standards of Nature Communications, and I recommend publication in its current form.